# LABEL PROPAGATION WITH WEAK SUPERVISION

**Rattana Pukdee**\*, **Dylan Sam**\*, **Maria-Florina Balcan, Pradeep Ravikumar**
Machine Learning Department
Carnegie Mellon University
Pittsburgh, USA
{rpukdee , dylansam}@cs.cmu.edu

## ABSTRACT

Semi-supervised learning and weakly supervised learning are important paradigms that aim to reduce the growing demand for labeled data in current machine learning applications. In this paper, we introduce a novel analysis of the classical label propagation algorithm (LPA) (Zhu & Ghahramani, 2002) that moreover takes advantage of useful prior information, specifically probabilistic hypothesized labels on the unlabeled data. We provide an error bound that exploits both the local geometric properties of the underlying graph and the quality of the prior information. We also propose a framework to incorporate *multiple* sources of noisy information. In particular, we consider the setting of weak supervision, where our sources of information are weak labelers. We demonstrate the ability of our approach on multiple benchmark weakly supervised classification tasks, showing improvements upon existing semi-supervised and weakly supervised methods.

## 1 INTRODUCTION

High-dimensional machine learning models require large labeled datasets for good performance and generalization. In the paradigm of semi-supervised learning, we look to overcome the bottleneck of labeled data by leveraging large amounts of unlabeled data and assumptions on how the target predictor behaves over the unlabeled samples. In this work, we focus on the classical semi-supervised approach of label propagation (LPA) (Zhu & Ghahramani, 2002; Zhou et al., 2003). This method propagates labels from labeled to unlabeled samples, under the assumption that the target predictor is smooth with respect to a graph over the samples (that is frequently defined by a euclidean distance threshold or nearest neighbors). However, in practice, to satisfy this strong assumption, the graph can be highly disconnected. In these cases, LPA performs well locally on regions connected to labeled points, but has low overall coverage as it cannot propagate to points beyond these connected regions.

In practice, we also have additional side-information beyond such smoothness of the target predictor. One concrete example of side information comes from the field of weakly supervised learning (WSL) (Ratner et al., 2016; 2017), which considers learning predictors from domain knowledge that takes the form of hand-engineered weak labelers. These weak labelers are heuristics that provide multiple weak labels per unlabeled sample, and the focus in WSL is to aggregate these weak labels to produce noisy pseudolabels for each unlabeled sample. In practice, weak labelers are typically not designed to be smooth with respect to a graph, even though the underlying target predictor might be. For example, weak labelers are commonly defined as hard, binary predictions, with an ability to abstain from predicting. We thus see that LPA and WSL have complementary sources of information, as smoothing via LPA can improve the quality of weak labelers. By encouraging smoothness, predictions near multiple abstentions can be made more uncertain, and abstentions can be converted into predictions by confident nearby predictions.

In this paper, we first bolster the theoretical foundations of LPA in the presence of side information. While LPA has a strong theoretical motivation of leveraging smoothness of the target predictor, there is limited theory on how accurate the propagated labels actually are. As a key contribution of this paper, we provide a "fine-grained" theory of LPA when used with any *general prior* on the target classes of the unlabeled samples. We provide a novel error bound for LPA, which depends on key

---

\*Equal contribution

local geometric properties of the graph, such as underlying smoothness of the target predictor over the graph, and the flow of edges from labeled points, as well as the accuracy of our prior. Our bound provides an intuition as to when LPA should prioritize propagating label information or when it should prioritize using prior information. We provide a comparison of our error bound to an existing spectral bound (Belkin & Niyogi, 2004) and demonstrate that our bound is preferable in some examples.

Next, we propose a framework for incorporating multiple sources of noisy information to LPA by extending a framework from Zhu et al. (2003). We construct additional "dongle" nodes in the graph that correspond to individual noisy labels. With these additional nodes, we connect them to unlabeled points that receive noisy predictions and perform label propagation on this new graph as usual. We study multiple different techniques for determining the weight on these additional edges.

Finally, we focus on the specific case when our side information comes from WSL. We provide experimental results on standard weakly supervised benchmark tasks (Zhang et al., 2021) to support our theoretical claims and to compare our methods to standard LPA, other semi-supervised methods, and existing weakly supervised baselines. Our experiments demonstrate that incorporating smoothness via LPA in the standard weakly supervised pipeline leads to better performance, outperforming many existing WSL algorithms. This supports that there are significant benefits to combining LPA and WSL, and we believe that this intersection is a fertile ground for future research.

## 1.1 RELATED WORK

**Label propagation** Many papers have studied LPA from a theoretical standpoint. LPA has various connections to random walks, spectral clustering (Zhu et al., 2003), manifold learning (Belkin & Niyogi, 2004; Belkin et al., 2006) and network generative models (Yamaguchi & Hayashi, 2017), graph conductance (Talukdar & Cohen, 2014). Another line of research in LPA proposes using prior information at the initialization of LPA (Yamaguchi et al., 2016; Zhou et al., 2018), with applications in image segmentation (Vernaza & Chandraker, 2017), distant supervision (Bing et al., 2015), and domain adaptation (Cai et al., 2021; Wei et al., 2020). Finally, as the graph has a large impact on the performance of LPA, another line of work studies how to optimize the construction of the graph with linear-based (Wang & Zhang, 2007) methods, manifold-based (Karasuyama & Mamitsuka, 2013) methods, or deep learning based methods (Liu et al., 2018; 2019).

**Weakly supervised learning** The field of (programmatic) weakly supervised learning provides a framework for creating and combining hand-engineered weak labelers (Ratner et al., 2016; 2017; 2019; Fu et al., 2020) to pseudolabel unlabeled data and train a downstream model. Recent advances in weakly supervised learning extend the setting to include a small set of labeled data. One recent line of work has considered constraining the space of possible pseudolabels via weak labeler accuracies (Arachie & Huang, 2019; Mazzetto et al., 2021a;b; Arachie & Huang, 2021; 2022). Other works improve the aggregation scheme (Xu et al., 2021) or the weak labelers (Awasthi et al., 2020). We note that only one method incorporates any notion of smoothness into the weakly supervised pipeline (Chen et al., 2022). This work leverages the smoothness of pretrained embeddings in clustering. While clustering and LPA have similar intuitions, they result in fundamentally different notions of smoothness. We also remark that this paper does not consider the semi-supervised setting.

**Semi-supervised learning** Many other methods in semi-supervised learning look to induce smoothness in a learnt model. These include consistency regularization (Bachman et al., 2014; Sajjadi et al., 2016; Samuli & Timo, 2017; Sohn et al., 2020) and co-training (Blum & Mitchell, 1998; Balcan et al., 2004; Han et al., 2018). In addition, Graph Neural Networks (GNNs) (Kipf & Welling, 2017; Hamilton et al., 2017; Gilmer et al., 2017; Scarselli et al., 2008; Gori et al., 2005; Henaff et al., 2015) is a class of deep learning based methods that also operate over graphs. Some recent works (Huang et al., 2020; Wang & Leskovec, 2020; Dong et al., 2021) have made connections between graph neural networks and LPA. While all these methods focus on a similar goal of learning a smooth function, they do not address the weakly supervised setting.

## 2 PRELIMINARIES

We consider a binary classification setting where we want to learn a classifier $f^* : \mathcal{X} \rightarrow \{0, 1\}$. We observe a small set of labeled data $L = \{(x_i, y_i)\}_{i=1}^n$ and a much larger set of unlabeled data $U = \{x_j\}_{j=n+1}^{n+m}$. LPA relies on the assumption that nearby data points have similar labels. This is

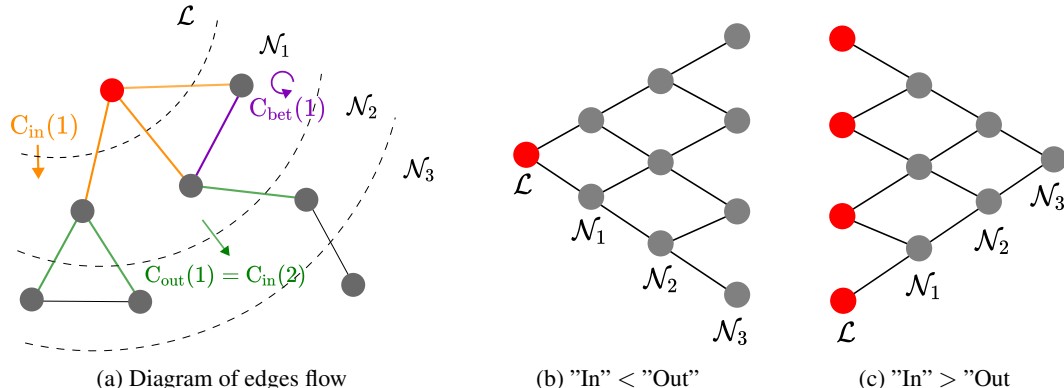

(a) Diagram of edges flow      (b) "In" < "Out"      (c) "In" > "Out

Figure 1: A diagram of edges flow between neighborhoods of $L$. Color on each edge implies that the edge contributes to which flows (In, Between, Out) (left). Examples of graphs with different structure, where colored points represent labeled points (middle, right).

expressed in terms of smoothness with respect to an undirected graph $G = (V, E)$, with $|V|$ nodes representing each point $x \in L \cup U$, and with an adjacency matrix $W = (w)_{ij}$. LPA then leverages the assumption that adjacent points in this graph have similar labels, by propagating label information from $L$ to $U$. Specifically, it learns $f : \mathcal{X} \to \mathbb{R}$ by solving the following optimization problem:

$$\min_{f \in \mathbb{R}^{n+m}} \frac{1}{2} \left( \sum_{i=1}^{n+m} \sum_{j=1}^{n+m} w_{ij}(f_i - f_j)^2 \right) \text{ s.t. } f_i = y_i \text{ for } i \leq n$$

where $f \in \mathbb{R}^{n+m}$ is the prediction vector and, abusing notation, $f_i = f(x_i)$. The method generalizes to the multi-class setting by replacing $y_i$ with a one-hot-encoding vector, and predicting a score vector at each node. Zhu et al. (2003) provides a quick iterative method to solve this optimization problem.

## 3 LABEL PROPAGATION WITH PRIOR INFORMATION

We analyze LPA with initial noisy predictions $h(x) : \mathcal{X} \to [0, 1]$, by solving the following objective:

$$\min_{f \in \mathbb{R}^{n+m}} \frac{1}{2} \left( \sum_{i=1}^{n+m} \sum_{j=1}^{n+m} w_{ij}(f_i - f_j)^2 + \mu \sum_{i=1}^{n+m} (f_i - h(x_i))^2 \right) \text{ s.t. } f_i = y_i \text{ for } i \leq n, \quad (1)$$

where $\mu \in \mathbb{R}$ determines how much the solution is regularized to be close to $h$. In the standard LPA, we have no prior information on the unlabeled points, which can be seen as the case when $h = 0.5$ and $\mu \to 0$. In our theory, $h$ can be any general prior.

### 3.1 ERROR BOUND OF LPA WITH PRIOR INFORMATION

Similar to the standard LPA, there exists a closed form optimal solution of Equation 1, which is discussed in Appendix A. We know that, for the optimal solution of Equation 1, we can bound the error of a point $i$ ($|f_i^* - y_i|$) by the error of its neighbors and terms corresponding to the smoothness of the true labels and the prior information accuracy; we formally state this in Appendix B (Lemma 2). Because the error on labeled points are zero, we can bound the error in terms of the distance of a point to the nearest labeled point. For a set of labeled data $L$, let $\mathcal{N}(L)$ be a set of reachable points where there is at least one path from a point in $L$. Define a set of neighbors $k$-hops away from $L$ as $\mathcal{N}_k(L)$ (i.e, a set of points whose shortest path to a point in $L$ is length $k$). Let $l$ be the number of hops required to cover $\mathcal{N}(L)$. Then, we have

$$\mathcal{N}(L) = L \cup \bigcup_{k=1}^{l} \mathcal{N}_k(L).$$

For simplicity, we denote $\mathcal{N}_k$ as $\mathcal{N}_k(L)$ and $\mathcal{N}_0$ as $L$. We now define terms that are fundamental to our error bound. First, we introduce notions of *In-flow*, *Between-flow*, and *Out-flow*, which represent the fraction of edges that flow in, between, and out of $\mathcal{N}_k(L)$.

**Definition 1.** *For a graph $G$ with an adjacency matrix $W = (w)_{ij}$ and a set of k-hop neighbors $\mathcal{N}_k$, we define the In-flow, Between-flow and Out-flow of $\mathcal{N}_k$ as*

$$C_{in}(k) = \sum_{i \in \mathcal{N}_k, j \in \mathcal{N}_{k-1}} w_{ij}, \quad C_{bet}(k) = \sum_{i \in \mathcal{N}_k, j \in \mathcal{N}_k} w_{ij},$$
$$C_{out}(k) = \sum_{i \in \mathcal{N}_k, j \in \mathcal{N}_{k+1}} w_{ij}$$

These terms are related to the notion of conductance, which measures the fraction of out-going edges from any subset of nodes. We can write the Dirichlet conductance (HaoChen et al., 2021) of a neighborhood $\mathcal{N}_k$ as follows

$$\phi(\mathcal{N}_k) = \frac{C_{\text{in}}(k) + C_{\text{out}}(k)}{C_{\text{in}}(k) + C_{\text{bet}}(k) + C_{\text{out}}(k)}.$$

**Definition 2.** *(Ratio between Out-flow and In-flow)*

$$\gamma_k = \frac{C_{out}(k)}{C_{in}(k) + \mu|\mathcal{N}_k|}$$

$\gamma_k$ is a proportion of the Out-flow and In-flow edges of a neighborhood (see Figure 1 for graphs with different flow). Next, we define the smoothness of $\mathcal{N}_k$, prior information error, and average error.

**Definition 3.** *(Smoothness of neighborhood) For $1 \le k \le l$, we define the smoothness of true labels of points in $\mathcal{N}_k$ with respect to the graph as*

$$s_k = \sum_{i \in \mathcal{N}_k} \sum_j w_{ij} |y_j - y_i|.$$

**Definition 4.** *(Prior information error) For $1 \le k \le l$, let the average error of the prior in $\mathcal{N}_k$ be*

$$\alpha_k = \frac{\sum_{i \in \mathcal{N}_k} |h_i - y_i|}{|\mathcal{N}_k|}.$$

**Definition 5.** *(Average error) We define the average error at the $\mathcal{N}_k$ as*

$$E_k = \frac{\sum_{i \in \mathcal{N}_k} |f_i^* - y_i|}{|\mathcal{N}_k|}.$$

**Theorem 1.** *(Informal version of Theorem 3) Let $f^*$ be the optimal solution of the optimization problem of Equation 1, under Assumption 1 which assume that average error of a fraction of points that has "Out" connections from neighborhood $\mathcal{N}_k$ is of a constant factor of the average error in $\mathcal{N}_k$ (refer to Appendix B), the error of $f^*$ in each neighborhood $\mathcal{N}_k$ is given by*

$$E_k \le O\left(\sum_{i=1}^k d_i\right),$$

*where*

$$d_k = \sum_{i=k}^l c_i \left(\prod_{j=k}^{i-1} \gamma_j\right), \quad c_k = \frac{s_k + \mu|\mathcal{N}_k|\alpha_k}{C_{in}(k) + \mu|\mathcal{N}_k|}.$$

*Proof.* (Sketch) The key idea of our proof is to upper bound each $E_i$ for $i \in \{1, \dots, l\}$ by exploiting the insight that we can bound the average error of a set $N_i$ (points that are $i$ hops away from labeled points) with the average errors of its neighbors $N_{i-1}$ and $N_{i+1}$ by using Lemma 2. We first bound $E_1$ with $E_0 = 0$ and $E_2$, then we bound $E_2$ with $E_1$ and $E_3$, and so on. See Appendix B for the full version of our proof. □

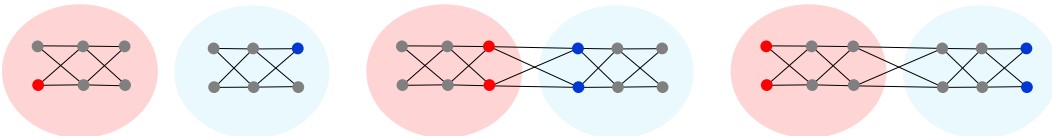

Figure 2: Example of graphs $G_1, G_2, G_3$ (left, mid, right) to compare our bound to existing bounds. The background color represents the true label class, and colored points represents labeled points.

$c_k$ is a combination of *smoothness* $s_k$ and the *prior accuracy* $\alpha_k$, and $\mu$ controls the trade-off between using information from the graph or the initialization. When $\mu = 0$, we recover the standard LPA without any prior. On the other hand, $\mu \to \infty$ is equivalent to only using the initial predictions.

$d_k$ is a linear combination of $c_i$ for $k \leq i \leq l$ where the coefficient of each $c_i$ is given by $\prod_{j=k}^{i-1} \gamma_j$, representing the influence from $\mathcal{N}_i$. When $\gamma_j < 1$ ("In" ¿ "Out"), the influence is exponentially small while when $\gamma_j > 1$ ("Out" ¿ "In") the influence can be exponentially large. This aligns with our intuition that when we have more "In" than "Out", we will have a better guarantee. We remark that if $c_k = 0$, regardless of the product $\prod_{j=k}^{i-1} \gamma_j$, $c_k$ will make no contribution to the bound.

To tighten this upper bound, we have to reduce both $c_k$ and $\gamma_k$. In doing so, the value of $\mu$ is important; a larger value of $\mu$ reduces both $c_k$ and $\gamma_k$ by increasing their denominators. Thus, given similar levels of smoothness and prior accuracy ($\frac{s_k}{C_{\text{in}}(k)} \approx \alpha_k$), it is better to use a larger value of $\mu$, that is we should rely more on the prior information.

The *number of hops* ($k$) from labeled points $L$ also plays a key role in the bound. The upper bound on $E_k$ is given by a linear combination of $k$ terms, so points that are closer to $L$ will have a smaller $k$ and a better guarantee. This encourages us to have a more connected graph, requiring fewer hops to reach all points. However, adding noisy edges may potentially decrease the smoothness of the graph.

## 3.2 COMPARISON WITH PRIOR (SPECTRAL) BOUNDS

We compare our bound with an existing bound that relies on spectral analysis (Belkin & Niyogi, 2004). This bound is for LPA with a soft constraint, given by the problem

$$\min_{f \in \mathbb{R}^{n+m}} \sum_{i=1}^{n+m} \sum_{j=1}^{n+m} w_{ij}(f_i - f_j)^2 + \eta \sum_{i \leq n} (f_i - y_i)^2. \tag{2}$$

We define the empirical error and generalization error as:

$$R_n(f) = \frac{1}{n} \sum_{i=1}^{n} (f_i - y_i)^2, \quad R(f) = \frac{1}{n+m} \sum_{i=1}^{n+m} (f_i - y_i)^2$$

As we do not have a hard constraint ($f_i = y_i$ for $i \leq n$), the empirical error is not necessary zero.

**Theorem 2.** *(Generalization performance of graph regularization (simplified version)) Let $f$ be the optimal solution of Equation 2, $n \geq 4$ be the number of randomly sampled labeled points from some graph $G$ and $\lambda_1$ be the second smallest eigenvalue of the Laplacian matrix of $G$. With probability $1 - \delta$, we have*

$$|R_n(f) - R(f)| \leq \beta + \sqrt{\frac{2 \log(2/\delta)}{n}} (n\beta + 4)$$

*where*

$$\beta = \frac{3\eta^2 \sqrt{n}}{(\lambda_1 - \eta)^2} + \frac{4\eta}{\lambda_1 - \eta}$$

The original version of this bound as in (Belkin & Niyogi, 2004) is in Appendix C. We consider graphs $G_1, G_2, G_3$ in Figure 2 to compare the bounds. Here $v(G)$ refers to the value of parameter $v$ for a graph $G$.

First, we note that this bounds the difference between the empirical error and the generalization error, while our bound is for the generalization error itself. The spectral bound assumes that we have

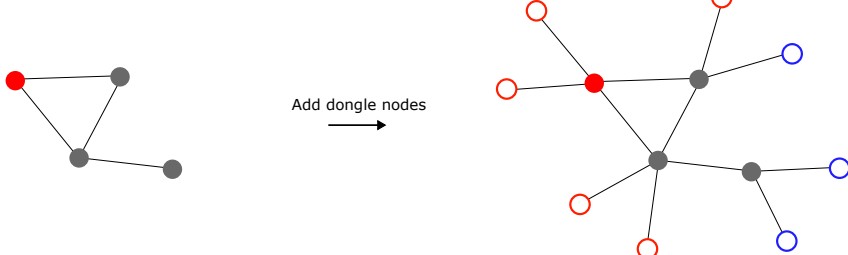

Figure 3: One can turn the label propagation with multiple sources of information into a standard label propagation problem by augmenting the graph $G$ (left) with dongle nodes (right). The colored point represents a labeled point. The points without the shade are dongle nodes.

randomly sampled initial labeled points, and thus the bound only depend on the number of labeled points. For example, $G_2$ and $G_3$ have the same underlying graph and the same number of labeled points, so they have the same spectral bound of generalization error, which relies on the empirical error. In contrast, our bound takes the position of labeled points into account to provide an explicit explanation why LPA performs better on $G_2$ than $G_3$. We can see this since $G_2$ is smoother than $G_3$ ($c_2(G_2) = 0, c_2(G_3) = \frac{s_2(G_3)}{C_{\text{in}}(2)(G_3)} = \frac{8}{8} = 1$).

The spectral bound depends on the second smallest eigenvalue $\lambda_1$. If the graph is not well clustered, $\lambda_1$ will be small. For example, $\lambda_1(G_1) = 2, \lambda_1(G_2) = 0.53$. Belkin & Niyogi (2004) suggests that when $\lambda_1$ is small, we should cut the graph in two, using the eigenvector corresponding to $\lambda_1$, and optimize the objective separately. Our bound works for any graph, in fact, our bound is also tight on $G_2$ where LPA achieves zero error (as $c_1(G_2) = c_2(G_2) = 0$). Also, as $\eta \to \infty$, the objective of Equation 2 is equivalent to Equation 1. The, the spectral bound takes on value $\beta \to 3\sqrt{n} - 4$, which implies that it does not depend on the geometry of the graph ($\lambda_1$) anymore. Finally, the spectral bound does not use any prior information, while our bound captures the interplay between the quality of graph and the quality of prior information.

## 4 LABEL PROPAGATION WITH MULTIPLE SOURCES OF INFORMATION

We now consider the setting where we observe multiple sources of prior information and provide a framework to incorporate them into LPA. Assume that we have multiple initial noisy predictions $h_i(x) : \mathcal{X} \to [0, 1]$ for $i = 1, 2, \ldots, k$. A natural extension of the LPA objective is given by

$$\sum_{i=1}^{n+m} \sum_{j=1}^{n+m} w_{ij}(f_i - f_j)^2 + \sum_{i=1}^{n+m} \sum_{j=1}^{k} (f_i - h_j(x_i))^2 \alpha_j(x_i) \qquad (3)$$

such that $f_i = y_i$ for $i \le n$. The first term encourages our prediction to be smooth with respect to a graph while the second term encourage our prediction to also be close to the initial predictions. The function $\alpha_j : \mathcal{X} \to [0, \infty)$, which we need to learn, controls how close we want our final prediction to be to each initial prediction $h_j$. We can turn this into a standard label propagation problem for which we have an efficient iterative method to solve by augmenting the graph $G$ with dongle nodes (Appendix D). Given a fixed $\alpha_j$ for each $j = 1, \ldots, k$, we can also show that there exists an initial prediction $h$ where the solution of Equation 3 is equivalent to a solution of LPA with a single initial prediction $h$ for which our analysis applies (Appendix E).

A key question is for this framework is "how to choose $\alpha_j$ ?" In the ideal setting, we set $\alpha_j(x_i) = 0$ when $h_j$ makes an incorrect prediction for point $x_i$ and set $\alpha_j(x_i) = 1$ when $h_j$ makes a correct prediction,

$$\alpha_j(x_i) = 1[1[h_j(x_i) > 0.5] = y_i].$$

However, this is not applicable in a practical setting as knowing when $h_j$ is correct or incorrect at a point $x_i$ is equivalent to knowing the corresponding true label $y_i$ of that point. We now investigate different approaches to select the function $\alpha_j$.

## 4.1 Estimated Accuracy

Although we do not know whether $h_j$ will make a correct prediction at each point $x_i$, we can still approximate its accuracy over the entire dataset. We can use techniques from crowd-sourcing literature or weak supervision literature to approximate the accuracy of each noisy labeler. Then, we can set $\alpha_j$ as the estimated accuracy of a $h_j$,

$$\alpha_j(x_i) = \mathbb{P}(1[h_j(x) > 0.5] = y) = p_j.$$

We also consider setting $\alpha_j = \ln(\frac{p_j}{1-p_j})$ as in the boosting literature (in Appendix F.1).

## 4.2 Probabilistic Approach

We can also consider a probabilistic approach to select the function $\alpha_j$. Let each $h_j$ be sampled from a Gaussian distribution

$$h_j \sim \mathcal{N}(y, \sigma_j(x)^2)$$

and let $f$ follow the Gaussian field as in Zhu & Ghahramani (2002), where

$$\rho_\beta(f) \propto \exp(-\beta E(f)), \quad E(f) = \frac{1}{2} \sum_{i=1}^{n+m} \sum_{j=1}^{n+m} w_{ij}(f_i - f_j)^2.$$

Then, the log-likelihood is given by

$$l(f) = \text{constant} - \sum_{i=1}^{n+m} \sum_{j=1}^{k} \frac{1}{2\sigma_j(x_i)^2}(h_j(x_i) - f_i)^2 - \frac{\beta}{2} \sum_{i=1}^{n+m} \sum_{j=1}^{n+m} w_{ij}(f_i - f_j)^2.$$

This resembles the objective of Equation 3 and suggests that we should set our function $\alpha_j$ as

$$\alpha_j(x_i) = \frac{1}{\sigma_j(x_i)^2},$$

where $\sigma_j(x_i)^2$ is the variance of $h_j$ at point $x_i$. With access to a small set of labeled data points, we can estimate $\sigma_j(x_i)$ through heteroscedastic regression (Wasserman, 2006), which is further discussed in Appendix F.2. We note that this function $\alpha_j$ changes over values of $x$ as it is computed through regression, while the accuracy-based weighting has a constant value for $\alpha_j$.

## 5 Experiments

We connect LPA and the field of weak supervision by using weak labelers as our source of prior information. Formally, a set of weak labelers is given by $\lambda = \{\lambda_1, ..., \lambda_k\}$, where each $\lambda_i : \mathcal{X} \to \{0, 1, \emptyset\}$ and $\emptyset$ denotes an abstention. Here, we consider LPA with a single source of prior information $h(x) = h_\lambda(x)$ is an aggregation of weak labelers, which we refer to as **LPA+WL**. For this method, we use Snorkel MeTaL (Ratner et al., 2019) as our aggregation scheme. We also consider our extensions of LPA with multiple sources of prior information when we set $h_i = \lambda_i$, for each weak labeler. We refer to our extensions of LPA that incorporate weak labelers through dongle nodes as **LPAD (A)** and **LPAD (P)**, where the last letter denotes our techniques to estimate the weighted edges of these dongle nodes (accuracy, and probabilistic approach). For methods that require accuracies, we use accuracies estimated via Snorkel MeTaL. We note that LPAD (A) and LPA+WL are both using the Snorkel estimated accuracy, we provide a discussion on their difference in Appendix E.1. For LPA + WL, we set $\mu = 1$. Further experimental details for our methods and the baselines are in Appendix G.

We compare our approaches to existing weak supervision methods, standard LPA, and other semi-supervised baselines on 4 binary classification datasets from the WRENCH benchmark (Zhang et al., 2021). The features from these text and image datasets are extracted from BERT (Kenton & Toutanova, 2019) and ResNet (He et al., 2016b) respectively. On each dataset, we balance the training data to have equal class proportions. To generate a small set of labeled data, we randomly sample $n = 100$ points from the training data. The remaining data serves as our unlabeled training data. For all graph-based methods, we construct a graph $G$ with average degree $t$, which is a hyperparameter of our method, and with edges that have value 1. More information about $t$ and other hyperparameters of all approaches are in Appendix G.2. Code to replicate our experiments can be found here[1].

---

[1]https://github.com/dsam99/label_propagation_weak_supervision

| Method | Youtube | SMS | Basketball | CDR |
|--------|---------|-----|------------|-----|
| LPA | $82.00 \pm 1.37$ | $94.32 \pm 0.45$ | $78.71 \pm 2.41$ | $67.41 \pm 0.82$ |
| GCN | $84.16 \pm 0.95$ | $94.32 \pm 1.02$ | $61.34 \pm 1.16$ | $65.42 \pm 1.00$ |
| Snorkel + L | $87.44 \pm 0.47$ | $96.24 \pm 0.32$ | $82.08 \pm 0.83$ | $68.03 \pm 0.28$ |
| FS + L | $87.76 \pm 0.51$ | $94.84 \pm 0.43$ | $70.23 \pm 1.20$ | $67.70 \pm 0.29$ |
| CLL | $88.56 \pm 0.80$ | $94.56 \pm 0.73$ | $77.02 \pm 3.96$ | $68.52 \pm 0.58$ |
| Liger + L | $88.72 \pm 0.58$ | $96.08 \pm 0.38$ | $80.98 \pm 1.71$ | $67.33 \pm 0.18$ |
| **LPA + WL** | $88.32 \pm 0.50$ | $96.80 \pm 0.36$ | $83.13 \pm 1.43$ | $67.61 \pm 0.19$ |
| **LPAD (A)** | $90.32 \pm 0.43$ | $96.32 \pm 0.52$ | $83.06 \pm 0.74$ | $68.13 \pm 0.74$ |
| **LPAD (P)** | $87.84 \pm 0.53$ | $96.64 \pm 0.39$ | $82.01 \pm 2.96$ | $68.97 \pm 0.51$ |

Table 1: We report accuracy ($\pm$ s.e.) on a held-out **test dataset** for training an endmodel on pseudolabeled training data, when averaged over 5 seeds. We highlight the best performing method in red and the second best performing method in blue.

## 5.1 BASELINES

We compare our methods against various semi-supervised and existing weakly supervised learning approaches. We use ground truth labels instead of pseudolabels on the 100 labeled points in methods denoted with (+L). In all these approaches, we train an inductive endmodel on the pseudolabeled training data, as is standard in WSL literature.

**Label Propagation (LPA):** The standard label propagation baseline (Zhu & Ghahramani, 2002) on graph $G$. This does not take into account weak labeler information.

**Graph Convolutional Network (GCN):** We provide results for a standard graph convolutional network (Kipf & Welling, 2017). This method also does not take into account weak labeler information.

**Snorkel + L:** A weakly supervised learning aggregation scheme, Snorkel MeTaL (Ratner et al., 2017; 2019), which produces pseudolabels through a graphical model.

**FlyingSquid + L (FS + L):** Another weakly supervised method that estimates parameters of a graphical model via a triplet method (Fu et al., 2020).

**Constrained Label Learning (CLL):** A method that produces an labeling contained within a feasible space constrained by the error rates of weak labelers (Arachie & Huang, 2021). We use the small set of labeled data to generate the error rates of the weak labelers.

**Liger + L:** A method that extends weak labelers using the smoothness of pretrained models and develops cluster-level aggregations (Chen et al., 2022). This method uses FS (Fu et al., 2020) as a base aggregation scheme.

## 5.2 RESULTS

We provide results for test accuracy of training an endmodel on pseudolabels generated by the baselines and our methods in Table 1. Our results demonstrate that incorporating weak labels into label propagation improves upon the performance of LPA across all datasets (LPA + WL > LPA). In addition, in almost every dataset, using LPA to incorporate smoothness improves upon the prior aggregation of weak labels (LPA + WL > Snorkel + L). Our methods also outperform other weakly supervised aggregation methods, in most cases. The best performing baseline we compare to is Liger or CLL, which each only marginally outperforms some of our methods on one dataset. We note that there is not clear best weighting scheme between (A) and (P), although they outperform most baselines on almost all tasks. In addition, one of our methods is the best performing approach on all datasets.

We also report the accuracy of standard LPA and our methods on the labeled and unlabeled training data in Table 2 (i.e, evaluating pseudolabel accuracies). We measure the accuracy of a particular

| Method | YouTube | | SMS | | CDR | |
|---|---|---|---|---|---|---|
| | Accuracy | Coverage | Accuracy | Coverage | Accuracy | Coverage |
| Snorkel + L | $75.96 \pm 0.13$ | $89.75 \pm 0.08$ | $70.40 \pm 0.34$ | $48.31 \pm 0.52$ | $70.64 \pm 0.12$ | $92.41 \pm 0.11$ |
| LPA | $55.98 \pm 0.08$ | $11.97 \pm 0.16$ | $54.71 \pm 0.01$ | $9.42 \pm 0.03$ | $50.79 \pm 0.00$ | $1.58 \pm 0.00$ |
| Liger + L | $81.06 \pm 0.47$ | $99.98 \pm 0.01$ | $\mathbf{78.62 \pm 0.23}$ | $96.01 \pm 0.20$ | $50.56 \pm 0.13$ | $81.79 \pm 10.72$ |
| **LPA + WL** | $76.02 \pm 0.12$ | $89.81 \pm 0.08$ | $70.75 \pm 0.35$ | $49.03 \pm 0.52$ | $70.64 \pm 0.12$ | $92.41 \pm 0.11$ |
| **LPAD (A)** | $84.03 \pm 0.15$ | $89.81 \pm 0.08$ | $70.80 \pm 0.26$ | $49.00 \pm 0.51$ | $\mathbf{72.87 \pm 0.08}$ | $91.66 \pm 0.49$ |
| **LPAD (P)** | $\mathbf{89.52 \pm 0.13}$ | $89.75 \pm 0.08$ | $70.98 \pm 0.27$ | $49.03 \pm 0.52$ | $71.91 \pm 0.25$ | $92.22 \pm 0.11$ |

Table 2: We report accuracy and coverage ($\pm$ s.e.) of the various label propagation methods on the full partially labeled **training data** (i.e, pseudolabel accuracies), when averaged over 5 seeds. We also add Liger + L as it looks to improve coverage by extending weak labelers. We bold the best performing method (in terms of Accuracy) on each dataset.

approach on abstained datapoints as $50\%$ (i.e, random guessing on binary data) on all abstained points as the method has no information on these points. Results for additional datasets are deferred to Appendix 3; on these datasets, the weak labeler coverage is almost 100% of the data, so coverage is roughly the same across our methods and the baselines. We observe that coverage drastically increases when using our weakly supervised prior (Table 2) over the standard LPA and slightly over that of Snorkel. We also observe that our method improves upon the base aggregation method of Snorkel on almost all datasets, improving both overall accuracy and coverage due to the propagation of information to nearby points. We note that Liger has much higher coverage on YouTube and SMS, although the accuracy on this larger set is much worse (see Table 3 in the Appendix).

## 6    DISCUSSION

We provide a novel theoretical perspective on LPA that takes advantage of useful prior information. Our bound differs significantly from existing spectral bounds, and provides insight into how to best incorporate priors into LPA. We note that our analysis is general and works with *any* initialization $h$. We also provide a framework to handle multiple sources of side information and empirical results for the setting of weak supervision. Further work can incorporate other types of prior information into LPA, such as in the recent line of work of learning with past predictions (Mitzenmacher & Vassilvitskii, 2021; Khodak et al., 2022). In addition, our connections of LPA with weakly supervised learning illustrate (both theoretically and empirically) that these methods benefit each other. As a whole, our results support adding smoothness to the standard WSL pipeline and can encourage further connections between semi-supervised learning algorithms and WSL. We note a few limitations of our method; our bound depends on several parameters, smoothness $s_k$, prior information accuracy $\alpha_k$; in general, we may need to approximate these values through labeled data. It remains an open question of how to do this effectively. In addition, we assume a uniformity assumption that the average error of points with "Out" connections from $\mathcal{N}_k$ is of a constant factor of the average error in $\mathcal{N}_k$. Relaxing the bound beyond this assumption is also an open question.

We remark that our approach bridges the gap between classical label propagation and modern deep learning by incorporating information from pretrained models to construct our graph $G$. Since we construct $G$ through Euclidean distance, our work uses notions of smoothness in the learnt embeddings, which is also noted in Chen et al. (2022). As we gain access to more powerful pretrained models, our approach will also benefit through a better graph $G$. This method also provides a natural framework to combine information from large pretrained models (via our graph $G$) and rules provided by domain experts (through our prior predictions $h_1, \ldots, h_k$).

ACKNOWLEDGEMENTS

This work was supported in part by NSF grants IIS-1909816, IIS-1955532, IIS-2211907, CCF-1910321 and DARPA under cooperative agreement HR00112020003 and funding from Bosch Center for Artificial Intelligence and the ARCS Foundation.

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

# A   CLOSED FORM SOLUTION OF LPA

We provide a closed form solution of the following optimization problem.

$$\min_{f \in \mathbb{R}^{n+m}} \frac{1}{2} (\sum_{i,j} w_{ij}(f_i - f_j)^2 + \mu||f - h||_2^2) \text{ s.t. } f_i = y_i \text{ for } i \leq n.$$

Here we abuse notation $h$ as a vector $(h(x_1), \ldots, h(x_{n+m}))$ and $h_i = h(x_i)$. For simplicity we refer $i \in L$ as $1 \leq i \leq n$ and $i \in U$ as $n + 1 \leq i \leq n + m$. First, note that

$$\frac{1}{2} (\sum_{i,j} w_{ij}(f_i - f_j)^2 = \frac{1}{2} (\sum_{i \in L} \sum_{j \in L} w_{ij}(f_i - f_j)^2 + 2 \sum_{i \in L} \sum_{j \in U} w_{ij}(f_i - f_j)^2$$
$$+ \sum_{i \in U} \sum_{j \in U} w_{ij}(f_i - f_j)^2)$$

The first term is a constant as $f_i = y_i$ for $i \in L$. Denote $\mathbf{f}_L \in \mathbb{R}^n$ is a column vector with entry $f_i$ for $i \in L$ and $\mathbf{f}_U \in \mathbb{R}^m$ is a column vector with entry $f_j$ for $j \in U$. We sometimes refer $w_{i,j}$ to $w_{ij}$. For the second term we have

$$\sum_{i \in L} \sum_{j \in U} w_{ij}(f_i - f_j)^2 = \sum_{i \in L} \sum_{j \in U} w_{ij}(f_i^2 - 2f_i f_j + f_j^2)$$
$$= \sum_{i \in L}(\sum_{j \in U} w_{ij})f_i^2 + \sum_{j \in U}(\sum_{i \in L} w_{ij})f_j^2 - 2\sum_{i \in L} \sum_{j \in U} f_i w_{ij} f_j$$
$$= \text{constant} + \mathbf{f}_U^T D_{UL} \mathbf{f}_U - 2\mathbf{f}_L^T W_{LU} \mathbf{f}_U.$$

where $D_{UL} \in \mathbb{R}^{m \times m}$ is a diagonal matrix with $(D_{UL})_{jj} = \sum_{i \in L} w_{i,j+n}$ and $W_{LU} \in \mathbb{R}^{n \times m}$ is a matrix with entry $(W_{LU})_{ij} = w_{i,j+n}$. For the third term, we have

$$\frac{1}{2} \sum_{i \in U} \sum_{j \in U} w_{ij}(f_i - f_j)^2 = \frac{1}{2} \sum_{i \in U} \sum_{j \in U} w_{ij}(f_i^2 - 2f_i f_j + f_j^2)$$
$$= \sum_{i \in U}(\sum_{j \in U} w_{ij})f_i^2 - \sum_{i \in U} \sum_{j \in U} f_i w_{ij} f_j$$
$$= \mathbf{f}_U^T D_{UU} \mathbf{f}_U - \mathbf{f}_U^T W_{UU} \mathbf{f}_U$$

where $D_{UU} \mathbb{R}^{m \times m}$ is a diagonal matrix with $(D_{UU})_{jj} = \sum_{i \in U} w_{i,j+n}$ and $W_{UU} \in \mathbb{R}^{m \times m}$ is a matrix with entry $(W_{UU})_{ij} = w_{i+n,j+n}$. Therefore, the overall objective is given by

$$\min_{\mathbf{f}_U \in \mathbb{R}^m} \text{constant} + \mathbf{f}_U^T (D_{UL} + D_{UU} - W_{UU}) \mathbf{f}_U - 2\mathbf{f}_L^T W_{LU} \mathbf{f}_U + \frac{\mu}{2} ||f - h||_2^2.$$

Differentiating with respect to $\mathbf{f}_U$ and setting equal to 0, we have

$$2(D_{UL} + D_{UU} - W_{UU})\mathbf{f}_U - 2W_{LU}^T \mathbf{f}_L + 2\mu(\mathbf{f}_U - \mathbf{h}_U) = 0$$
$$\mathbf{f}_U = (D_{UL} + D_{UU} - W_{UU} + \mu I_d)^{-1}(\mu \mathbf{h}_U + W_{LU}^T \mathbf{f}_L)$$

when $\mathbf{h}_U \in \mathbb{R}^m$ with $(\mathbf{h}_U)_j = h_{j+n}$. We can also extend this to the case when $\mu \in \mathbb{R}^{m+n}$, where we have a different value of $\mu_i$ for each $i$. The optimization objective is given by

$$\min_{f \in \mathbb{R}^{n+m}} \frac{1}{2}(\sum_{i,j} w_{ij}(f_i - f_j)^2 + \sum_i \mu_i(f_i - h_i)^2) \text{ s.t. } f_i = y_i \text{ for } i \leq n.$$

We can write the regularization term for $U$ as

$$\sum_{i \in U} \mu_i(f_i - h_i)^2 = (\mathbf{f}_U - \mathbf{h}_U)^T D_\mu (\mathbf{f}_U - \mathbf{h}_U)$$

when $D_\mu \in \mathbb{R}^m$ is a diagonal matrix with entry $(D_\mu)_{jj} = \mu_{j+n}$. We can write the optimization objective as

$$\min_{\mathbf{f}_U \in \mathbb{R}^m} \text{constant} + \mathbf{f}_U^T (D_{UL} + D_{UU} - W_{UU}) \mathbf{f}_U - 2\mathbf{f}_L^T W_{LU} \mathbf{f}_U + (\mathbf{f}_U - \mathbf{h}_U)^T D_\mu (\mathbf{f}_U - \mathbf{h}_U).$$

Differentiating with respect to $\mathbf{f}_U$ and setting equal to 0, we have

$$2(D_{UL} + D_{UU} - W_{UU})\mathbf{f}_U - 2W_{LU}^T \mathbf{f}_L + 2D_\mu(\mathbf{f}_U - \mathbf{h}_U) = 0$$
$$\mathbf{f}_U = (D_{UL} + D_{UU} + D_\mu - W_{UU})^{-1}(D_\mu \mathbf{h}_U + W_{LU}^T \mathbf{f}_L)$$

## B  THEORETICAL RESULTS

First, we analyze the closed form solution of the LPA in Lemma 1.

**Lemma 1.** *Let $f^*$ be the optimal solution of the optimization problem of Equation 1 then for $n + 1 \leq i \leq n + m$,*

$$f_i^* = \frac{\sum_j w_{ij} f_j^* + \mu h_i}{\sum_j w_{ij} + \mu}$$

*Proof.* For each $i \leq n$, we must have $f_i^* = y_i$ to satisfy the hard constraints. For each $n + 1 \leq i \leq n + m$, we differentiate the objective with respect to $f_i$ and set equal to 0, resulting in

$$\sum_j w_{ij}(f_i - f_j) + \mu(f_i - h_i) = 0.$$

Rearranging this and setting $f_j = f_j^*$, we have the lemma. □

Next, we will analyze the error $|f_i^* - y_i|$, which is the difference between the optimal solution and the true label. Note that $|f_i^* - y_i| < 0.5$ implies that we have a correct soft label $f_i^*$.

**Lemma 2.** *Let $f^*$ be the optimal solution of the optimization problem of Equation 1. Then, for $n + 1 \leq i \leq n + m$, we have*

$$|f_i^* - y_i| \leq \frac{\sum_j w_{ij}|f_j^* - y_j| + \sum_j w_{ij}|y_j - y_i| + \mu|h_i - y_i|}{\sum_j w_{ij} + \mu}.$$

*Proof.* From lemma 1,

$$
\begin{aligned}
|f_i^* - y_i| &= \left| \frac{\sum_j w_{ij} f_j^* + \mu h_i}{\sum_j w_{ij} + \mu} - y_i \right| \\
&= \left| \frac{\sum_j w_{ij}(f_j^* - y_i) + \mu(h_i - y_i)}{\sum_j w_{ij} + \mu} \right| \\
&= \left| \frac{\sum_j w_{ij}(f_j^* - y_j) + \sum_j w_{ij}(y_j - y_i) + \mu(h_i - y_i)}{\sum_j w_{ij} + \mu} \right| \\
&\leq \frac{\sum_j w_{ij}|f_j^* - y_j| + \sum_j w_{ij}|y_j - y_i| + \mu|h_i - y_i|}{\sum_j w_{ij} + \mu}
\end{aligned}
$$

□

Lemma 2 says that we can bound the error of point $i$, $|f_i^* - y_i|$ by the error of its neighbors $|f_j^* - y_j|$ and terms corresponding to the smoothness of the true labels and the prior information accuracy. Because we know that the error on labeled points are zero, this lemma motivates us to bound the error in term of the distance of our points from the labeled points.

Next, in addition to the average error defined in Definition 5, we define the In-error, Between-error and Out-error for each $\mathcal{N}_k$.

**Definition 6.** *For a graph $G$ with an adjacency matrix $W = (w)_{ij}$, a set of k-hop neighbors $\mathcal{N}_k$ and a prediction $f \in \mathbb{R}^{n+m}$, we define the In-error, Between-error and Out-error of $\mathcal{N}_k$ as*

$$Err_{in}(f, y, k) = \frac{\sum_{i \in \mathcal{N}_k, j \in \mathcal{N}_{k-1}} w_{ij}|f_i - y_i|}{C_{in}(k)}$$

$$Err_{bet}(f, y, k) = \frac{\sum_{i \in \mathcal{N}_k, j \in \mathcal{N}_k} w_{ij}|f_i - y_i|}{C_{bet}(k)}$$

$$Err_{out}(f, y, k) = \frac{\sum_{i \in \mathcal{N}_k, j \in \mathcal{N}_{k+1}} w_{ij}|f_i - y_i|}{C_{out}(k)}$$

*For simplicity, we will write $E_{in}(k)$, $E_{bet}(k)$, $E_{out}(k)$ for $Err_{in}(f^*, y, k)$, $Err_{bet}(f^*, y, k)$, $Err_{out}(f^*, y, k)$ respectively.*

We will use Lemma 2, to derive a relationship between errors in $\mathcal{N}_k$ and its neighbors. We make use of the fact that the Out-flow of $\mathcal{N}_k$ is the same as the In-flow of $\mathcal{N}_{k+1}$.

**Lemma 3.** *For* $0 \leq k \leq l-1$

$$C_{out}(k) = C_{in}(k+1)$$

**Lemma 4.** *(Error difference inequality) For* $1 \leq k \leq l-1$*, we have*

$$C_{in}(k)(E_{in}(k) - E_{out}(k-1)) + \sum_{i \in \mathcal{N}_k} \mu|f_i^* - y_i| \leq C_{out}(k)(E_{in}(k+1) - E_{out}(k)) + s_k + \mu|\mathcal{N}_k|\alpha_k$$

*where* $s_k$ *is the smoothness of true labels and* $\alpha_k$ *is the prior information error over* $\mathcal{N}_k$*.*

*Proof.* From lemma 2, we have

$$\sum_j w_{ij}|f_i^* - y_i| + \mu|f_i^* - y_i| \leq \sum_j w_{ij}|f_j^* - y_j| + \sum_j w_{ij}|y_j - y_i| + \mu|h_i - y_i|.$$

We take a summation over $i \in \mathcal{N}_k$,

$$\sum_{i \in \mathcal{N}_k} \sum_j w_{ij}|f_i^* - y_i| + \mu|f_i^* - y_i| \leq \sum_{i \in \mathcal{N}_k} (\sum_j w_{ij}|f_j^* - y_j| + \sum_j w_{ij}|y_j - y_i| + \mu|h_i - y_i|).$$

From the definition of the In-error, Between-error, Out-error, smoothness $s_k$, and weak label error $\alpha_k$, we have

$$\text{LHS} = C_{\text{in}}(k)\text{E}_{\text{in}}(k) + C_{\text{bet}}(k)\text{E}_{\text{bet}}(k) + C_{\text{out}}(k)\text{E}_{\text{out}}(k) + \sum_{i \in \mathcal{N}_k} \mu|f_i^* - y_i|$$

$$\text{RHS} = C_{\text{out}}(k-1)\text{E}_{\text{out}}(k-1) + C_{\text{bet}}(k)\text{E}_{\text{bet}}(k) + C_{\text{in}}(k+1)\text{E}_{\text{in}}(k+1) + s_k + \mu|\mathcal{N}_k|\alpha_k.$$

From lemma 3, we know that

$$C_{\text{out}}(k) = C_{\text{in}}(k+1), \tag{4}$$

so we can rearrange the inequality as

$$C_{\text{in}}(k)(\text{E}_{\text{in}}(k) - \text{E}_{\text{out}}(k-1)) + \sum_{i \in \mathcal{N}_k} \mu|f_i^* - y_i| \leq C_{\text{out}}(k)(\text{E}_{\text{in}}(k+1) - \text{E}_{\text{out}}(k)) + s_k + \mu|\mathcal{N}_k|\alpha_k.$$

$\square$

**Lemma 5.** *(Error difference inequality* $k = l$*)*

$$C_{in}(l)(E_{in}(l) - E_{out}(l-1)) + \sum_{i \in \mathcal{N}_l} \mu|f_i^* - y_i| \leq s_l + \mu|\mathcal{N}_l|\alpha_l$$

*Proof.* Similar to lemma 4, we have

$$\sum_{i \in \mathcal{N}_l} \sum_j w_{ij}|f_i^* - y_i| + \mu|f_i^* - y_i| \leq \sum_{i \in \mathcal{N}_l} (\sum_j w_{ij}|f_j^* - y_j| + \sum_j w_{ij}|y_j - y_i| + \mu|h_i - y_i|)$$

Because, $\mathcal{N}_l$ is the last neighborhood, there is no edge out from $\mathcal{N}_l$ and

$$\text{LHS} = C_{\text{in}}(l)\text{E}_{\text{in}}(l) + C_{\text{bet}}(l)\text{E}_{\text{bet}}(l) + \sum_{i \in \mathcal{N}_l} \mu|f_i^* - y_i|$$

$$\text{RHS} = C_{\text{out}}(l-1)\text{E}_{\text{out}}(l-1) + C_{\text{bet}}(l)\text{E}_{\text{bet}}(l) + s_l + \mu|\mathcal{N}_l|\alpha_l$$

and rearrange to

$$C_{\text{in}}(l)(\text{E}_{\text{in}}(l) - \text{E}_{\text{out}}(l-1)) + \sum_{i \in \mathcal{N}_l} \mu|f_i^* - y_i| \leq s_l + \mu|\mathcal{N}_l|\alpha_l.$$

$\square$

We can see that this inequality contains different notions of error. We now define the proportion between In-error and Out-error.

**Definition 7.** *Let $a_k, b_k$ be the proportion of the In-error and Out-error with the average error,*

$$a_k = \frac{E_{in}(k)}{E_k}, b_k = \frac{E_{in}(k)}{E_k}.$$

*when*

$$E_k = \frac{\sum_{i \in \mathcal{N}_k} |f_i^* - y_i|}{|\mathcal{N}_k|}.$$

**Assumption 1.** *(Uniformity of error) We assume that the In-error and Out-error are roughly the same as the average error in each neighborhood.*

$$a_k = O(1), b_k = O(1), \frac{b_k}{a_k} = O(1)$$

For example, any graph $G$ that has all points in a neighborhood $\mathcal{N}_k$ with the same number of edges that go into and out from that point, has the property that $a_k = b_k = 1$. In particular, assume that we have 2 points in $\mathcal{N}_k$, the first point has 4 edges from $\mathcal{N}_{k-1}$ and 2 edges to $\mathcal{N}_{k+1}$ while the second point has 2 edges from $\mathcal{N}_{k-1}$ and 1 edge to $\mathcal{N}_{k+1}$, this graph still has $a_k = b_k = 1$. In general, we expect the proportion $\frac{b_k}{a_k}$ to be close to 1. Next, we will substitute $a_k, b_k$ in Lemma 4.

**Corollary 1.** *For $1 \leq k \leq l - 1$, we have*

$$(a_k E_k - b_{k-1} E_{k-1}) \leq \frac{C_{out}(k)}{C_{in}(k) + \mu|\mathcal{N}_k|}(a_{k+1} E_{k+1} - b_k E_k) + \frac{s_k + \mu|\mathcal{N}_k|\alpha_k}{C_{in}(k) + \mu|\mathcal{N}_k|}$$

*Proof.* From lemma 4

$$C_{\text{in}}(k)(\text{E}_{\text{in}}(k) - \text{E}_{\text{out}}(k-1)) + \sum_{i \in \mathcal{N}_k} \mu|f_i^* - y_i| \leq C_{\text{out}}(k)(\text{E}_{\text{in}}(k+1) - \text{E}_{\text{out}}(k)) + s_k + \mu|\mathcal{N}_k|\alpha_k$$

We let $\text{E}_{\text{in}}(k) = a_k \text{E}_k$ and $\text{E}_{\text{out}}(k) = b_k \text{E}_k$ and $\sum_{i \in \mathcal{N}_k} |f_i^* - y_i| = |\mathcal{N}_k| \text{E}_k$.

$$C_{\text{in}}(k)(a_k \text{E}_k - b_{k-1} \text{E}_{k-1}) + \mu|\mathcal{N}_k|\text{E}_k \leq C_{\text{out}}(k)(a_{k+1} \text{E}_{k+1} - b_k \text{E}_k) + s_k + \mu|\mathcal{N}_k|\alpha_k$$
$$C_{\text{in}}(k)(a_k \text{E}_k - b_{k-1} \text{E}_{k-1}) + \mu|\mathcal{N}_k|(\text{E}_k - \text{E}_{k-1}) \leq C_{\text{out}}(k)(a_{k+1} \text{E}_{k+1} - b_k \text{E}_k) + s_k + \mu|\mathcal{N}_k|\alpha_k,$$

as we know that $E_{k-1} \geq 0$. Then, simplifying yields that

$$(C_{\text{in}}(k) + \mu|\mathcal{N}_k|)(a_k \text{E}_k - b_{k-1} \text{E}_{k-1}) \leq C_{\text{out}}(k)(a_{k+1} \text{E}_{k+1} - b_k \text{E}_k)) + s_k + \mu|\mathcal{N}_k|\alpha_k$$
$$(a_k \text{E}_k - b_{k-1} \text{E}_{k-1}) \leq \frac{C_{\text{out}}(k)}{C_{\text{in}}(k) + \mu|\mathcal{N}_k|}(a_{k+1} \text{E}_{k+1} - b_k \text{E}_k) + \frac{s_k + \mu|\mathcal{N}_k|\alpha_k}{C_{\text{in}}(k) + \mu|\mathcal{N}_k|}.$$

$\square$

**Corollary 2.** *We have*

$$(a_l E_l - b_{l-1} E_{l-1}) \leq \frac{s_l + \mu|\mathcal{N}_l|\alpha_l}{C_{in}(l) + \mu|\mathcal{N}_l|}.$$

The corollary implies that the difference between the error between neighborhood can't be too large. We introduce the next two lemma to help deriving the bound.

**Lemma 6.** *For $d_1, d_2, \ldots, d_l$ that satisfies the following inequalities,*

$$d_k \leq \gamma_k d_{k+1} + c_k$$

*for $1 \leq k \leq l - 1$ and*

$$d_l \leq c_l.$$

*We have*

$$d_k \leq \sum_{i=k}^{l} c_i \left(\prod_{j=k}^{i-1} \gamma_j\right)$$

*Proof.* The main idea is that we can use the upper bound on $d_l, d_{l-1}, \ldots, d_{k+1}$ to find the upper bound of $d_k$. First, we start with $d_{l-1}$

$$d_{l-1} \leq \gamma_{l-1} d_l + c_{l-1}$$
$$\leq \gamma_{l-1} c_l + c_{l-1}.$$

Next, we continue with $d_{l-2}$,

$$d_{l-2} \leq \gamma_{l-2} d_{l-1} + c_{l-2}$$
$$\leq \gamma_{l-2}(\gamma_{l-1} c_l + c_{l-1}) + c_{l-2}.$$
$$= c_l \gamma_{l-1}\gamma_{l-2} + c_{l-1}\gamma_{l-2} + c_{l-2}$$

and so on. With this idea, we can show by induction that

$$d_k \leq \sum_{i=k}^{l} c_i (\prod_{j=k}^{i-1} \gamma_j).$$

We sum these inequalities up to have the lemma. □

**Lemma 7.** *For* $x_1, x_2, \ldots, x_l$ *that satisfies the following inequalities,*

$$a_k x_k - b_{k-1} x_{k-1} \leq d_k$$

*for* $1 \leq k \leq l$, *when* $a_k, b_k, d_k$ *are positive constant. We have*

$$x_k \leq \frac{1}{a_k}(\sum_{i=1}^{k} d_i (\prod_{j=i}^{k-1} \delta_j)) + \frac{a_1}{a_k}(\prod_{j=1}^{k-1} \delta_j) x_0$$

*when*

$$\delta_j = \frac{b_j}{a_j}$$

*Proof.* We divide both side of the inequality by $a_k$, for each $1 \leq k \leq l$, we have

$$x_k \leq \frac{b_{k-1}}{a_k} x_{k-1} + \frac{d_k}{a_k}.$$

We can recursively apply this inequality,

$$x_k \leq \frac{b_{k-1}}{a_k}(\frac{b_{k-2}}{a_{k-1}} x_{k-2} + \frac{d_{k-1}}{a_{k-1}}) + \frac{d_k}{a_k}.$$
$$= \frac{1}{a_k}(\frac{b_{k-1}}{a_{k-1}} b_{k-2} x_{k-2} + \frac{b_{k-1}}{a_{k-1}} d_{k-1} + d_k)$$
$$= \frac{1}{a_k}(\delta_{k-1} b_{k-2} x_{k-2} + \delta_{k-1} d_{k-1} + d_k)$$
$$\leq \frac{1}{a_k}(\delta_{k-1} b_{k-2}(\frac{b_{k-3}}{a_{k-2}} x_{k-3} + \frac{d_{k-2}}{a_{k-2}}) + \delta_{k-1} d_{k-1} + d_k)$$
$$\leq \frac{1}{a_k}(\delta_{k-1}\delta_{k-2} b_{k-3} x_{k-3} + \delta_{k-1}\delta_{k-2} d_{k-2} + \delta_{k-1} d_{k-1} + d_k)$$
$$\leq \ldots$$
$$\leq \frac{1}{a_k}(\sum_{i=1}^{k} d_i (\prod_{j=i}^{k-1} \delta_j)) + \frac{a_1}{a_k}(\prod_{j=1}^{k-1} \delta_j) x_0$$

□

Now, we are ready to derive the error bound of LPA.

**Theorem 3.** *Let $f^*$ be the optimal solution of the optimization problem of Equation 1, the error of $f^*$ in each neighborhood is given by*

$$E_k \le \frac{1}{a_k}(\sum_{i=1}^{k} d_i(\prod_{j=i}^{k-1} \delta_j))$$

*when*

$$\delta_j = \frac{b_j}{a_j}, \quad d_k = \sum_{i=k}^{l} c_i(\prod_{j=k}^{i-1} \gamma_j)$$

*and*

$$c_k = \frac{s_k + \mu|\mathcal{N}_k|\alpha_k}{C_{in}(k) + \mu|\mathcal{N}_k|}, \quad \gamma_k = \frac{C_{out}(k)}{C_{in}(k) + \mu|\mathcal{N}_k|}.$$

*Under assumption 1, we have*

$$E_k \le O(\sum_{i=1}^{k} d_i)$$

*Proof.* From corollary 1, 2 we have

$$(a_k \mathrm{E}_k - b_{k-1}\mathrm{E}_{k-1}) \le \frac{C_{\text{out}}(k)}{C_{\text{in}}(k) + \mu|\mathcal{N}_k|}(a_{k+1}\mathrm{E}_{k+1} - b_k\mathrm{E}_k) + \frac{s_k + \mu|\mathcal{N}_k|\alpha_k}{C_{\text{in}}(k) + \mu|\mathcal{N}_k|}$$

and

$$(a_l \mathrm{E}_l - b_{l-1}\mathrm{E}_{l-1}) \le \frac{s_l + \mu|\mathcal{N}_l|\alpha_l}{C_{\text{in}}(l) + \mu|\mathcal{N}_l|}.$$

Let

$$d_k = a_k\mathrm{E}_k - b_{k-1}\mathrm{E}_{k-1}, \quad c_k = \frac{s_k + \mu|\mathcal{N}_k|\alpha_k}{C_{\text{in}}(k) + \mu|\mathcal{N}_k|}, \quad \gamma_k = \frac{C_{\text{out}}(k)}{C_{\text{in}}(k) + \mu|\mathcal{N}_k|}$$

By lemma 6, we have

$$a_k\mathrm{E}_k - b_{k-1}\mathrm{E}_{k-1} = d_k \le \sum_{i=k}^{l} c_i(\prod_{j=k}^{i-1} \gamma_j).$$

By lemma 7, we have

$$E_k \le \frac{1}{a_k}(\sum_{i=1}^{k} d_i(\prod_{j=i}^{k-1} \delta_j)) + \frac{a_1}{a_k}(\prod_{j=1}^{k-1} \delta_j)E_0$$

$$= \frac{1}{a_k}(\sum_{i=1}^{k} d_i(\prod_{j=i}^{k-1} \delta_j))$$

when

$$\delta_j = \frac{b_j}{a_j}.$$

The last equality is true because the error $E_0 = 0$. With the assumption 1, $\frac{b_k}{a_k} = O(1)$, we have

$$E_k \le O(\sum_{i=1}^{k} d_i)$$

$\square$

## C  SPECTRAL BOUND

The following is the original version of the spectral generalization bound found in Belkin & Niyogi (2004), where they assume that we can have repeated labeled points (at most $u$ times).

**Theorem 4.** *(Generalization performance of graph regularization) Let $f$ be the optimal solution of Equation 2, $n \geq 4$ be the number of randomly sampled labeled points from some distribution where each vertex occurs no more than $u$ times, together with values $y_1, \ldots, y_n, |y_i| \leq M$. Let $\lambda_1$ be the second smallest eigenvalue of the Laplacian matrix of $G$. Assuming that $\forall \mathbf{x} |f(\mathbf{x})| \leq K$, we have with probability $1 - \delta$, (conditional on the multiplicity being no greater than $t$),*

$$|R_n(f) - R(f)| \leq \beta + \sqrt{\frac{2 \log(2/\delta)}{n}} \left( n\beta + (K + M)^2 \right)$$

*where*

$$\beta = \frac{3\eta^2 \sqrt{un}}{(\lambda_1 - \eta u)^2} + \frac{4\eta M}{\lambda_1 - \eta u}$$

We can set $u = 1, M = 1, K = 1$ to achieve the simplified version (Theorem 2).

## D  DONGLE NODES

We can change a label propagation problem with multiple initial predictions into a standard label propagation by augmenting a graph with dongle nodes (Zhu et al., 2003). Without loss of generality, we assume that each initial prediction has 3 possible outputs $h_j : \mathcal{X} \to \{\emptyset, 0, 1\}$ for $j = 1, \ldots, k$. However, this method also works for a general case when $h_j : \mathcal{X} \to [0, 1]$. We augment the original graph $G$ with the following nodes and edges,

1. For each weak labeler $h_j$, we add 2 nodes to the graph $G$ with vertices $v_{n+m+j}, v_{n+m+k+j}$. This represents a prediction of class 0 or 1 from the weak labeler $j$.
2. For each $x_i$ that $h_j(x_i) = 0$, we draw a weighted edge between $v_i$ (the corresponding vertex of $x_i$) and $v_{n+m+j}$ with weight $\alpha_j(x_i)$.
3. For each $x_i$ that $h_j(x_i) = 1$, we draw a weighted edge between $v_i$ (the corresponding vertex of $x_i$) and $v_{n+m+k+j}$ with weight $\alpha_j(x_i)$.
4. For each $x_i$ that $h_j(x_i) = \emptyset$, we do not draw any edge.

Let $G'$ be the new graph with a weighted adjacency matrix $(w'_{ij})$ then solving the objective of Equation 3 is equivalent to solving

$$\min_{f \in \mathbb{R}^{n+m+2k}} \sum_{i=1}^{n+m+2k} \sum_{j=1}^{n+m+2k} w'_{ij}(f_i - f_j)^2 \tag{5}$$

such that

1. $f_i = y_i$ for $i \leq n$.
2. $f_i = 0$ for $n + m + 1 \leq i \leq n + m + k$.
3. $f_i = 1$ for $n + m + k + 1 \leq i \leq n + m + 2k$.

We see initial predictions as dongle nodes and encode the parameter $\alpha_j(x_i)$ as a weight of an edge connecting the corresponding dongle node of predictor $j$ to the node of $x_i$. With a direct calculation, we can see that the objective of Equation 5 is the same as the original objective,

$$\sum_{i=1}^{n+m} \sum_{j=1}^{n+m} w_{ij}(f_i - f_j)^2 + \sum_{i=1}^{n+m} \sum_{j=1}^{k} (f_i - h_j(x_i))^2 \alpha_j(x_i)$$

such that $f_i = y_i$ for $i \leq n$. With this procedure, we add $2k$ nodes and at most $(n + m)k$ edges to $G$.

# E  CONNECTION BETWEEN LPA WITH MULTIPLE AND SINGLE INITIAL PREDICTIONS

We analyze the closed form solution of the optimization objective of Equation 3. By differentiating with respect to $f_i$, we know that the optimal solution $f_i^*$ satisfies the following

$$\sum_{j=1}^{n+m} 2w_{ij}(f_i^* - f_j^*) + \sum_{j=1}^{k} 2(f_i^* - h_j(x_i))\alpha_j(x_i) = 0$$

$$f_i^* = \frac{\sum_{j=1}^{n+m} w_{ij}f_j^* + \sum_{j=1}^{k} \alpha_j(x_i)h_j(x_i)}{\sum_{j=1}^{n+m} w_{ij} + \sum_{j=1}^{k} \alpha_j(x_i)}$$

From Lemma 1, recall that the optimal solution of LPA with an initial prediction (objective of Equation 1) is given by

$$f_i^* = \frac{\sum_j w_{ij}f_j^* + \mu h(x_i)}{\sum_j w_{ij} + \mu}$$

We can see that if we set

$$h(x_i) = \frac{\sum_{j=1}^{k} \alpha_j(x_i)h_j(x_i)}{\sum_{j=1}^{k} \alpha_j(x_i)}, \mu(x_i) = \sum_{j=1}^{k} \alpha_j(x_i),$$

the solution LPA with multiple initial predictions is equivalent to LPA with the initial prediction $h$, which could be seen as a weighted average prediction. The objective is given by

$$\min_{f \in \mathbb{R}^{n+m}} \frac{1}{2}\left(\sum_{i,j} w_{ij}(f_i - f_j)^2 + \sum_{i=1}^{n+m} \mu(x_i)(f_i - h(x_i))^2\right) \text{ s.t. } f_i = y_i \text{ for } i \leq n.$$

We note that now the parameter $\mu$ is now depends on each instance $x_i$. When we have

$$\sum_{j=1}^{k} \alpha_j(x_i) = \mu$$

is a constant for all $x_i$ then we will have the same setting as in the objective of Equation 1. However, our analysis still works in this case when $\mu(x_i)$ is not a constant.

## E.1  DIFFERENCE BETWEEN LPA+WL AND LPAD(A)

We note that LPAD (A) uses Snorkel to estimated accuracies $\alpha_j$. From above, LPAD (A) is equivalent to LPA with an initial prediction

$$h(x_i) = \frac{\sum_{j=1}^{k} \alpha_j h_j(x_i)}{\sum_{j=1}^{k} \alpha_j}, \quad \mu(x_i) = \sum_{j=1}^{k} \alpha_j.$$

We observe that $h$ is exactly the prior information for LPA+WL. However, the key difference is that for LPA + WL, we have a fixed $\mu$ for all data points, while in LPAD(A), the value $\mu(x_i)$ depends on $x_i$. To illustrate this, we consider 2 scenarios. First, we assume that we have 3 weak labelers $h_1, h_2$, and $h_3$, all with estimated accuracy 0.8. We consider a point $x_1$ with $h_1(x_1) = 1, h_2(x_1) = 1, h_3(x_1) = 1$ and $x_2$ with $h_1(x_2) = 1, h_2(x_2) = \emptyset, h_3(x_2) = \emptyset$, where $\emptyset$ is abstention. We can observe that

1. $h(x_1) = h(x_2) = 1$
2. $\mu(x_1) = 2.4, \ \mu(x_2) = 0.8$

Here in LPA + WL, $x_1, x_2$ have the same prior information and regularization parameter $\mu$. In LPAD(A), we put much more weight on the regularization parameter $\mu(x_1)$ than $\mu(x_2)$. This is intuitive as we should be more confident about our prior information when a higher number of weak labelers agree.

# F  METHODS FOR SELECTING ALPHA

## F.1  BOOSTING APPROACH

From boosting literature (Freund & Schapire, 1997), given many weak learners $h_j : \mathcal{X} \rightarrow \{0, 1\}$ for $j = 1, 2, \ldots, k$, an optimal way to combine these weak learner (corresponding to an exponential loss upper bound) is a weighted average

$$h = \frac{\sum_{j=1}^{k} \alpha_j h_j}{\sum_{j=1}^{k} \alpha_j}, \quad \alpha_j = \ln(\frac{\mathbb{P}(h_j(x) = y)}{1 - \mathbb{P}(h_j(x) = y)}),$$

Instead of accuracy, we could set $\alpha_j$ in this fashion suggested by the boosting literature. We show that this value of $\alpha_j$ minimizes the upper bound on the error $|f_i^* - y_i|$. Recall that the optimal solution of Equation 3 satisfies

$$f_i^* = \frac{\sum_{j=1}^{n+m} w_{ij} f_j^* + \sum_{j=1}^{k} \alpha_j(x_i) h_j(x_i)}{\sum_{j=1}^{n+m} w_{ij} + \sum_{j=1}^{k} \alpha_j(x_i)}$$

We can bound the error of a point $i$, $|f_i^* - y_i|$ by the error of its neighbor $|f_j^* - y_j|$. Observe that

$$f_i^* - y_i = \frac{\sum_{j=1}^{n+m} w_{ij} f_j^* + \sum_{j=1}^{k} \alpha_j(x_i) h_j(x_i)}{\sum_{j=1}^{n+m} w_{ij} + \sum_{j=1}^{k} \alpha_j(x_i)} - y_i$$

$$f_i^* - y_i = \frac{\sum_{j=1}^{n+m} w_{ij} (f_j^* - y_j + y_j - y_i) + \sum_{j=1}^{k} \alpha_j(x_i)(h_j(x_i) - y_i)}{\sum_{j=1}^{n+m} w_{ij} + \sum_{j=1}^{k} \alpha_j(x_i)}$$

$$|f_i^* - y_i| \leq \frac{\sum_{j=1}^{n+m} w_{ij} |f_j^* - y_j| + \sum_{j=1}^{n+m} w_{ij} |y_j - y_i| + |\sum_{j=1}^{k} \alpha_j(x_i)(h_j(x_i) - y_i)|}{\sum_{j=1}^{n+m} w_{ij} + \sum_{j=1}^{k} \alpha_j(x_i)}..$$

The first term represents errors of neighbor points $|f_j^* - y_j|$ and the second term represents the smoothness of the true labels on the graph $G$ and the third term represents the accuracy of the weighted prediction. We can improve the upper bound by selecting appropriate value of $\alpha_j(x_i)$ and $w_{ij}$ to minimize

$$|\frac{\sum_{j=1}^{k} \alpha_j(x_i)(h_j(x_i) - y_i)}{\sum_{j=1}^{k} \alpha_j(x_i)}| \geq |\frac{\sum_{j=1}^{k} \alpha_j(x_i)(h_j(x_i) - y_i)}{\sum_{j=1}^{n+m} w_{ij} + \sum_{j=1}^{k} \alpha_j(x_i)}|.$$

Consider the following lemma,

**Lemma 8.** *Given $k$ classifier $h_i : \mathcal{X} \rightarrow \{-1, 1\}$ for $i = 1, \ldots, k$. Let $h(x) = \sum_{i=1}^{k} \alpha_i h_i(x)$ be the weighted average among the classifiers. Assume that the prediction of $h_i(x)$ are independent between different $i$ . The optimal $\alpha_i$ that minimize the risk when the loss is exponential loss of $h(x)$,*

$$\mathcal{L}(h, x, y) = \exp(-yh(x))$$

*is given by*

$$\alpha_i = \frac{1}{2} \ln(\frac{\mathbb{P}(h_i(x) = y)}{1 - \mathbb{P}(h_i(x) = y)})$$

*Proof.* The risk is given by

$$\mathbb{E}(\mathcal{L}(h, x, y)) = \mathbb{E}(\exp(-yh(x)))$$

$$= \mathbb{E}(\exp(-y \sum_{i=1}^{k} \alpha_i h_i(x)))$$

$$= \prod_{i=1}^{k} \mathbb{E}(\exp(-y\alpha_i h_i(x)))$$

$$= \prod_{i=1}^{k} p_i \exp(-\alpha_i) + (1 - p_i) \exp(\alpha_i)$$

when $p_i = \mathbb{P}(h_i(x) = y)$. It is sufficient to choose $\alpha_i$ that maximize

$$p_i \exp(-\alpha_i) + (1 - p_i) \exp(\alpha_i).$$

Differentiate with respect to $\alpha_i$ and set to zero, we have

$$-p_i \exp(-\alpha_i) + (1 - p_i) \exp(\alpha_i) = 0$$
$$(1 - p_i) \exp(\alpha_i) = p_i \exp(-\alpha_i)$$
$$exp(2\alpha_i) = \frac{p_i}{1 - p_i}$$
$$\alpha_i = \frac{1}{2} \ln(\frac{p_i}{1 - p_i})$$

$\square$

Note that exponential loss is an upper bound of our hinge loss and Lemma 8 suggests that to minimize the exponential loss upper bound, we should set

$$\alpha_i = \frac{1}{2} \ln(\frac{p_i}{1 - p_i})$$

### F.2 HETEROSCEDASTIC REGRESSION

Recall that we model

$$h_j \sim \mathcal{N}(y, \sigma(x)^2)$$

so that we can write

$$h_j(x_i) = y(x_i) + \sigma_j(x_i)\varepsilon$$

when $\varepsilon \sim \mathcal{N}(0, 1)$. We want to regress $\sigma(x)$. Rearraging we have,

$$h_j(x_i) - y(x_i) = \sigma_j(x_i)\varepsilon$$
$$(h_j(x_i) - y(x_i))^2 = \sigma_j(x_i)^2 \varepsilon^2$$
$$\log((h_j(x_i) - y(x_i))^2) = \log(\sigma_j(x_i)^2) + \log(\varepsilon^2)$$

On labeled data, we can regress a function $g_j(x_i)$ to match $\log((h_j(x_i) - y(x_i))^2)$ then we set

$$\alpha_j(x_i) = \frac{1}{\exp(g_j(x_i))}.$$

## G ADDITIONAL EXPERIMENTAL DETAILS

We use the default splits from the WRENCH benchmark (Zhang et al., 2021) for each of our binary classification dataset. This benchmark has a Apache-2.0 license. For each text classification dataset (Youtube, SMS, CDR), we use pretrained BERT embeddings (Kenton & Toutanova, 2019). For our image classification tasks (Basketball), we use pretrained ResNet embeddings (He et al., 2016a). For each task, we balance the datasets and randomly sample 100 labeled datapoints. We balance the datasets to make sure that the overall sample of labeled data contains roughly the same amount of points from each class.

We use cluster compute resources to produce our empirical results. We use a single GPU (NVIDIA GeForce RTX 2080Ti) to run our methods and each of the baselines.

### G.1 WEAK LABEL SOURCES

We use the standard weak labels contained within the WRENCH benchmark (Zhang et al., 2021). These are standard in programmatic weak supervision literature and primarily consist of simple hand-engineered rules. For example, on the YouTube dataset (or a spam classification task), examples of weak labels are functions that check for the presence of words in a sentence (Figure 4). We defer interested readers to the benchmark (Zhang et al., 2021) and other papers in weak supervision (Ratner et al., 2017) for more details.

```
from snorkel.labeling import labeling_function

@labeling_function()
def check(x):
    return SPAM if "check" in x.text.lower() else ABSTAIN

@labeling_function()
def check_out(x):
    return SPAM if "check out" in x.text.lower() else ABSTAIN
```

Figure 4: Examples of weak labels on the YouTube dataset

## G.2 HYPERPARAMETER OPTIMIZATION

We perform hyperparameter optimization of all methods, selecting the best set of parameters on the validation set. We optimize over the following parameters for all methods' endmodels:

- learning rate: [0.01, 0.001, 0.0001]
- number of epochs: [20, 30, 40, 50]
- weight decay: [0, 0.01, 0.001]

In each experiment, we have a fixed batch size of 100 and a fixed architecture of a 2 layer neural network with a hidden dimension of 64 and a ReLU activation function. For all graph-based methods, we have an additional parameter $t \in [1, 2, 5, 10, 100]$. $t$ controls the average degree of nodes in $G$. Let $N$ be the number of nodes in $G$, we use the value $\frac{t}{N}$ and as our threshold percentile for our euclidean distance threshold graph. In essence, we add an edge between two points when the Euclidean distance between them is less than the $\frac{t}{N}$-th percentile of all $N^2$ pairwise distances. The motivation for this is that $N$ node corresponds to $N^2$ edges, so adding $\frac{t}{N}$ edges leads to a resulting graph with average degree $t$.

For our GCN baseline, we construct a graph $G$ in the same manner as all other LPA-based methods. The GCN architecture is a 2 layer neural network with hidden dimension of 16 and ReLU activations. Consequently, we train an endmodel on the pseudolabeled data, which is the same architecture as all other methods. For our Liger + L baseline, we optimizer over a fixed threshold value for their cosine similarity as some $k$ for each weak labeler. We note that this baseline is highly sensitive to the value of $k$; for BERT embeddings, we select values of $k \in [0.995, 0.9975, 1]$ as points are much less distinguished in the embedding space in comparison the larger foundation models (GPT-3, CLIP) in the original paper (Chen et al., 2022). We use 2 clusters for all tasks.

## H COMPLETE VERSION OF TABLES

We present our results for both training/pseudolabel performance (Table 3) and test/endmodel performance (Table 4) in more detail and with additional comparisons.

In our pseudolabel performance table, we provide the Non-Abstain accuracy (i.e, only considering accuracy on points on which the model makes a vote). We define abstaining as having a maximum logit (across either class) that is within $\epsilon = 0.001$ of 0.5. We observe a fundamental tradeoff: balancing high accuracy and little coverage against lower accuracy and higher coverage. We also observe that LPA performs well locally on regions connected to labeled points with non-abstain accuracy close to 100 percents, but has low overall coverage.

For our endmodel results, we additionally compare against a fully supervised approach that uses all of the training data and their labels. We remark that some datasets in the WRENCH benchmark have noisy labels, leading to imperfect fully supervised performance. For example, we also add evaluations on the Tennis dataset, which only achieves 88% fully supervised performance, and all methods seem to match this performance. We also add some additional variations of our dongle-based approach. We add a comparison to a boosting (Appendix F.1) to determine $\alpha$, which we refer to as LPAD (B). We also compare against a method that uses the unosberved ground truth accuracies for $\alpha$ (LPAD (O)) and another method that sets $\alpha = 1, \forall x$ (LPAD (1)). We note that LPAD (O) is an unfair comparison

| Method | YouTube | | | SMS | | | CDR | | |
|---|---|---|---|---|---|---|---|---|---|
| | Acc | Cov | NA Acc | Acc | Cov | NA Acc | Acc | Cov | NA Acc |
| Snorkel + L | $75.96 \pm 0.13$ | $89.75 \pm 0.08$ | $78.93 \pm 0.14$ | $70.40 \pm 0.34$ | $48.31 \pm 0.52$ | $92.20 \pm 0.29$ | $70.64 \pm 0.12$ | $92.41 \pm 0.11$ | $72.33 \pm 0.15$ |
| LPA | $55.98 \pm 0.08$ | $11.97 \pm 0.16$ | $100.00 \pm 0.00$ | $54.71 \pm 0.01$ | $9.42 \pm 0.03$ | $100.00 \pm 0.00$ | $50.79 \pm 0.00$ | $1.58 \pm 0.00$ | $100.00 \pm 0.00$ |
| Liger + L | $81.06 \pm 0.47$ | $99.98 \pm 0.01$ | $81.07 \pm 0.47$ | $78.62 \pm 0.23$ | $96.01 \pm 0.20$ | $79.81 \pm 0.18$ | $50.56 \pm 0.13$ | $81.79 \pm 10.72$ | $50.98 \pm 0.46$ |
| **LPA + WL** | $76.02 \pm 0.12$ | $89.81 \pm 0.08$ | $78.97 \pm 0.14$ | $70.75 \pm 0.35$ | $49.03 \pm 0.52$ | $92.32 \pm 0.29$ | $70.64 \pm 0.12$ | $92.41 \pm 0.11$ | $72.33 \pm 0.15$ |
| **LPAD (A)** | $84.03 \pm 0.15$ | $89.81 \pm 0.08$ | $87.89 \pm 0.16$ | $70.80 \pm 0.26$ | $49.00 \pm 0.51$ | $92.45 \pm 0.10$ | $\mathbf{72.87 \pm 0.08}$ | $91.66 \pm 0.49$ | $74.95 \pm 0.17$ |
| **LPAD (P)** | $\mathbf{89.52 \pm 0.13}$ | $89.75 \pm 0.08$ | $94.03 \pm 0.11$ | $\mathbf{70.98 \pm 0.27}$ | $49.03 \pm 0.52$ | $92.79 \pm 0.13$ | $71.91 \pm 0.25$ | $92.22 \pm 0.11$ | $73.75 \pm 0.25$ |
| **LPAD (B)** | $75.87 \pm 0.14$ | $89.81 \pm 0.08$ | $78.80 \pm 0.16$ | $70.83 \pm 0.24$ | $48.93 \pm 0.05$ | $92.58 \pm 0.16$ | $70.40 \pm 0.08$ | $91.64 \pm 0.50$ | $72.27 \pm 0.13$ |
| **LPAD (O)** | $89.36 \pm 0.08$ | $89.81 \pm 0.08$ | $93.8 \pm 0.10$ | $71.52 \pm 0.30$ | $49.00 \pm 0.51$ | $93.91 \pm 0.19$ | $73.91 \pm 0.07$ | $92.23 \pm 0.09$ | $75.93 \pm 0.09$ |
| **LPAD (1)** | $83.11 \pm 0.07$ | $76.97 \pm 0.11$ | $93.03 \pm 0.07$ | $70.88 \pm 0.26$ | $48.64 \pm 0.51$ | $92.93 \pm 0.09$ | $71.39 \pm 0.07$ | $75.00 \pm 0.11$ | $78.52 \pm 0.09$ |

| Method | Basketball | | | Tennis | | |
|---|---|---|---|---|---|---|
| | Acc | Cov | NA Acc | Acc | Cov | NA Acc |
| Snorkel + L | $69.09 \pm 0.02$ | $100.00 \pm 0.0$ | $69.09 \pm 0.02$ | $86.10 \pm 0.06$ | $100.00 \pm 0.0$ | $86.10 \pm 0.06$ |
| LPA | $62.16 \pm 0.02$ | $25.82 \pm 0.05$ | $97.12 \pm 0.17$ | $70.60 \pm 0.29$ | $52.61 \pm 0.34$ | $89.16 \pm 0.44$ |
| Liger + L | $59.06 \pm 2.53$ | $55.83 \pm 9.88$ | $65.21 \pm 1.17$ | $83.60 \pm 1.34$ | $100.00 \pm 0.00$ | $83.60 \pm 1.34$ |
| **LPA + WL** | $74.09 \pm 0.30$ | $99.94 \pm 0.04$ | $74.11 \pm 0.30$ | $86.91 \pm 0.14$ | $99.87 \pm 0.03$ | $86.96 \pm 0.15$ |
| **LPAD (A)** | $69.75 \pm 0.33$ | $99.95 \pm 0.03$ | $69.76 \pm 0.33$ | $87.15 \pm 0.05$ | $99.94 \pm 0.03$ | $87.17 \pm 0.06$ |
| **LPAD (P)** | $82.46 \pm 0.19$ | $90.42 \pm 0.32$ | $85.89 \pm 0.19$ | $87.69 \pm 0.11$ | $99.98 \pm 0.01$ | $87.70 \pm 0.12$ |
| **LPAD (B)** | $74.26 \pm 0.30$ | $99.95 \pm 0.02$ | $74.28 \pm 0.30$ | $87.20 \pm 0.08$ | $99.97 \pm 0.01$ | $87.21 \pm 0.08$ |
| **LPAD (O)** | $74.02 \pm 0.33$ | $99.99 \pm 0.01$ | $74.03 \pm 0.33$ | $87.23 \pm 0.09$ | $100.00 \pm 0.00$ | $87.23 \pm 0.09$ |
| **LPAD (1)** | $75.87 \pm 0.20$ | $69.97 \pm 0.23$ | $86.97 \pm 0.24$ | $87.46 \pm 0.17$ | $99.38 \pm 0.04$ | $87.70 \pm 0.17$ |

Table 3: We report accuracy, coverage, and non-abstaining accuracy (NA Acc) of the baselines and our variants of LPA on the **training data** (i.e, pseudolabel statistics), when averaged over 5 seeds.

| Method | Youtube | SMS | Basketball | CDR | Tennis |
|---|---|---|---|---|---|
| Snorkel + L | $87.44 \pm 0.47$ | $96.24 \pm 0.32$ | $82.08 \pm 0.83$ | $68.03 \pm 0.28$ | $88.51 \pm 0.04$ |
| FS + L | $87.76 \pm 0.51$ | $94.84 \pm 0.43$ | $70.23 \pm 1.20$ | $67.70 \pm 0.29$ | $88.56 \pm 0.02$ |
| CLL | $88.56 \pm 0.80$ | $94.56 \pm 0.73$ | $77.02 \pm 3.96$ | $68.52 \pm 0.58$ | $88.78 \pm 0.13$ |
| LPA | $82.00 \pm 1.37$ | $94.32 \pm 0.45$ | $78.71 \pm 2.41$ | $67.41 \pm 0.82$ | $83.35 \pm 3.30$ |
| GCN | $84.16 \pm 0.95$ | $94.32 \pm 1.02$ | $61.34 \pm 1.16$ | $65.42 \pm 1.00$ | $88.63 \pm 0.14$ |
| Liger + L | $88.72 \pm 0.58$ | $96.08 \pm 0.38$ | $80.98 \pm 1.71$ | $67.33 \pm 0.18$ | $86.43 \pm 0.87$ |
| **LPA + WL** | $88.32 \pm 0.50$ | $96.80 \pm 0.36$ | $83.13 \pm 1.43$ | $67.61 \pm 0.19$ | $88.51 \pm 0.04$ |
| **LPAD (A)** | $90.32 \pm 0.43$ | $96.32 \pm 0.52$ | $83.06 \pm 0.74$ | $68.13 \pm 0.74$ | $88.56 \pm 0.02$ |
| **LPAD (P)** | $87.84 \pm 0.53$ | $96.64 \pm 0.39$ | $82.01 \pm 2.96$ | $68.97 \pm 0.51$ | $88.60 \pm 0.04$ |
| **LPAD (B)** | $88.64 \pm 0.37$ | $96.56 \pm 0.33$ | $76.58 \pm 2.20$ | $67.01 \pm 0.43$ | $88.56 \pm 0.02$ |
| **LPAD (O)** | $90.16 \pm 0.50$ | $96.40 \pm 0.50$ | $81.10 \pm 1.43$ | $69.06 \pm 0.59$ | $88.58 \pm 0.02$ |
| **LPAD (1)** | $83.20 \pm 1.15$ | $94.32 \pm 0.35$ | $78.61 \pm 1.95$ | $67.76 \pm 0.21$ | $88.43 \pm 0.16$ |
| Fully Supervised | $89.92 \pm 1.45$ | $98.04 \pm 0.38$ | $86.04 \pm 2.02$ | $73.71 \pm 0.85$ | $88.43 \pm 1.06$ |

Table 4: We report accuracy on **test data** for training an endmodel on pseudolabeled training data, when averaged over 5 seeds. We bold baselines when they outperform both of our methods. We bold our methods when they outperform all baselines.

to all other methods as it accesses ground truth accuracies that other methods do not use; we add this comparison to describe the best potential performance of LPAD.

## I    HYPERPARAMETER ABLATION

We provide an ablation study on hyperparameter $t$. We report accuracy on the test data of an endmodel that is trained on pseudolabels from baselines and our methods with particular values of $t$ to determine the construction of $G$. We observe that all graph-based methods are sensitive to the choice of $t$, which controls the sparsity of edges in the (Euclidean) graph $G$. We remark that this finding is intuitive as most graph-based semi-supervised algorithms leverage properties of this graph to achieve better

performance. We can see a common trend among all methods where when $t$ is large, the end-model accuracy tends to decrease. We note that Snorkel + L does not leverage any graph information, but we still add it here for comparison.

| Method | YouTube | | | | |
| --- | --- | --- | --- | --- | --- |
| | $t = 1$ | $t = 2$ | $t = 5$ | $t = 10$ | $t = 100$ |
| Snorkel + L | $87.04 \pm 0.47$ | $85.44 \pm 0.9$ | $86.4 \pm 0.78$ | $86.4 \pm 0.78$ | $\mathbf{87.04 \pm 0.47}$ |
| LPA | $79.76 \pm 1.99$ | $81.6 \pm 1.15$ | $82.0 \pm 1.37$ | $82.64 \pm 1.62$ | $75.68 \pm 1.51$ |
| **LPA + WL** | $86.24 \pm 0.75$ | $86.56 \pm 0.81$ | $86.16 \pm 0.71$ | $88.32 \pm 0.5$ | $83.12 \pm 1.2$ |
| **LPAD (A)** | $89.12 \pm 0.5$ | $\mathbf{88.96 \pm 1.04}$ | $89.04 \pm 0.68$ | $\mathbf{90.32 \pm 0.43}$ | $84.0 \pm 0.54$ |
| **LPAD (B)** | $87.2 \pm 0.55$ | $86.24 \pm 1.22$ | $87.12 \pm 0.69$ | $88.64 \pm 0.37$ | $83.84 \pm 0.27$ |
| **LPAD (P)** | $\mathbf{87.76 \pm 0.79}$ | $87.84 \pm 0.53$ | $\mathbf{89.2 \pm 0.31}$ | $89.92 \pm 0.53$ | $82.8 \pm 1.03$ |

Table 5: We report accuracy on **test data** for training an endmodel on pseudolabeled training data from various label propagation methods when using different hyperparameter $t$ and averaged over 5 seeds.

| Method | SMS | | | | |
| --- | --- | --- | --- | --- | --- |
| | $t = 1$ | $t = 2$ | $t = 5$ | $t = 10$ | $t = 100$ |
| Snorkel + L | $95.04 \pm 0.46$ | $96.08 \pm 0.48$ | $96.24 \pm 0.32$ | $\mathbf{96.24 \pm 0.32}$ | $\mathbf{95.04 \pm 0.46}$ |
| LPA | $94.32 \pm 0.45$ | $94.52 \pm 0.26$ | $95.24 \pm 0.38$ | $92.2 \pm 1.08$ | $80.76 \pm 3.51$ |
| **LPA + WL** | $94.88 \pm 0.67$ | $96.44 \pm 0.26$ | $\mathbf{96.8 \pm 0.36}$ | $95.36 \pm 0.6$ | $85.12 \pm 3.82$ |
| **LPAD (A)** | $95.6 \pm 0.28$ | $\mathbf{96.8 \pm 0.33}$ | $96.32 \pm 0.52$ | $95.96 \pm 0.41$ | $86.12 \pm 4.27$ |
| **LPAD (B)** | $\mathbf{96.04 \pm 0.19}$ | $96.68 \pm 0.32$ | $96.56 \pm 0.33$ | $96.12 \pm 0.34$ | $82.64 \pm 4.36$ |
| **LPAD (P)** | $95.8 \pm 0.46$ | $95.64 \pm 0.36$ | $96.64 \pm 0.39$ | $96.16 \pm 0.48$ | $84.32 \pm 2.82$ |

Table 6: We report accuracy on **test data** for training an endmodel on pseudolabeled training data from various label propagation methods when using different hyperparameter $t$ and averaged over 5 seeds.

| Method | CDR | | | | |
| --- | --- | --- | --- | --- | --- |
| | $t = 1$ | $t = 2$ | $t = 5$ | $t = 10$ | $t = 100$ |
| Snorkel + L | $67.87 \pm 0.25$ | $\mathbf{68.03 \pm 0.28}$ | $68.24 \pm 0.72$ | $\mathbf{68.24 \pm 0.72}$ | $\mathbf{67.87 \pm 0.25}$ |
| LPA | $67.01 \pm 0.82$ | $65.17 \pm 1.09$ | $63.19 \pm 1.18$ | $59.33 \pm 1.8$ | $44.39 \pm 2.69$ |
| **LPA + WL** | $67.87 \pm 0.25$ | $67.61 \pm 0.19$ | $67.16 \pm 0.84$ | $65.8 \pm 0.82$ | $49.66 \pm 1.8$ |
| **LPAD (A)** | $68.13 \pm 0.74$ | $67.68 \pm 1.1$ | $\mathbf{68.65 \pm 0.53}$ | $67.56 \pm 0.38$ | $61.28 \pm 3.21$ |
| **LPAD (B)** | $66.44 \pm 1.23$ | $67.01 \pm 0.43$ | $65.98 \pm 0.92$ | $66.26 \pm 1.15$ | $54.06 \pm 2.78$ |
| **LPAD (P)** | $\mathbf{68.97 \pm 0.51}$ | $67.27 \pm 1.05$ | $65.48 \pm 0.36$ | $64.93 \pm 1.8$ | $58.34 \pm 3.71$ |

Table 7: We report accuracy on **test data** for training an endmodel on pseudolabeled training data from various label propagation methods when using different hyperparameter $t$ and averaged over 5 seeds.

| Method | Basketball | | | | |
| --- | --- | --- | --- | --- | --- |
| | $t = 1$ | $t = 2$ | $t = 5$ | $t = 10$ | $t = 100$ |
| Snorkel + L | $81.62 \pm 1.07$ | $\mathbf{83.62 \pm 0.8}$ | $\mathbf{82.08 \pm 0.83}$ | $82.08 \pm 0.83$ | $\mathbf{81.62 \pm 1.07}$ |
| LPA | $75.56 \pm 0.55$ | $79.36 \pm 1.49$ | $75.12 \pm 1.26$ | $78.71 \pm 2.41$ | $66.02 \pm 1.19$ |
| **LPA + WL** | $\mathbf{83.13 \pm 1.43}$ | $80.87 \pm 0.64$ | $79.95 \pm 1.56$ | $80.11 \pm 1.75$ | $73.45 \pm 1.09$ |
| **LPAD (A)** | $80.44 \pm 0.83$ | $78.9 \pm 1.53$ | $81.64 \pm 1.35$ | $\mathbf{83.06 \pm 0.75}$ | $74.83 \pm 1.49$ |
| **LPAD (B)** | $75.81 \pm 3.47$ | $72.75 \pm 2.11$ | $76.58 \pm 2.2$ | $78.0 \pm 4.34$ | $69.36 \pm 1.03$ |
| **LPAD (P)** | $82.01 \pm 2.96$ | $74.6 \pm 2.05$ | $73.31 \pm 5.25$ | $68.71 \pm 4.41$ | $67.18 \pm 3.64$ |

Table 8: We report accuracy on **test data** for training an endmodel on pseudolabeled training data from various label propagation methods when using different hyperparameter $t$ and averaged over 5 seeds.

