# OpenReview forum: "Label Propagation with Weak Supervision"
_ICLR.cc/2023/Conference — ICLR 2023 poster_

### Official Review · Reviewer_xkhm · 2022-10-22

**Confidence:** 4
**Correctness:** 4
**Technical Novelty And Significance:** 3
**Empirical Novelty And Significance:** 3
**Recommendation:** 8

**Clarity, Quality, Novelty And Reproducibility:**

The paper has new theoretical results and is technically sound. The authors also provide the codes for reproducibility. Overall, the paper is well-written.

**Strength And Weaknesses:**

Strengths:
1. The paper proposes a new error bound for LPA, and provides an example to show its superiority compared with conventional spectral bound.
2. The paper is technically sound.
3. The experimental results are good. The authors also provide their codes of experiments.

Weaknesses and questions:
1. I'm a little confused about its weaklly supervised learning setting. To my knowledge, there are many kinds of weaklly supervised learning setting. For exmple, in some settings, only coarse-grained labels are given for the training data; in some settings, the given labels are sometimes not correct for the training data. What is the specific setting of this paper? From the experiments, it seems that the labels (or psuedolabels) of some data are not correct? If so, what is the difference between it and learning with noisy data? And maybe it needs to compare with such learning methods with noisy data.
2. The paper compares its bound with spectral bound and gives a case study to show its superiority, which is good. Since the form of the two bounds are quite different, I think it's hard to say which is tighter in general. It may be interesting if you could provide the cases when the proposed bounds fail or at least worse than the conventional one. It can tell the readers when should we use the proposed bound and when should not.
3. Why does Table 2 not report the results of GCN, CLL and so on?

**Summary Of The Paper:**

This paper provides a new error bound of LPA with prior information and compares it with the conventional spectral bound. Then it proposes two strategies to design the prior information. The paper also conducts experiments, which compares with some weakly supervised methods and semi-supervised methods. The experimental results demonstrate the effectiveness of the proposed strategies.

**Summary Of The Review:**

Due to the strengths listed above, I recommend for Accept.

---

> ### Author Response · Authors · 2022-11-09
> **Response to Reviewer xkhm**
>
> Thank you for your review. We are pleased that you found our theoretical analysis to be “novel” and appreciate the “effectiveness” of our approach on various datasets. We hope the following discussion addresses the questions you had.
>
> > I'm a little confused about its weakly supervised learning setting
>
> Our weakly supervised learning setting is the same that is considered in (Ratner et al., 2016). Here, we have multiple sets of noisy labels. We have added an example of these noisy labels in Appendix G.1.
>
> > What is the difference between it and learning with noisy data? And maybe it needs to compare with such learning methods with noisy data.
>
> Yes, this is similar to learning with noisy data. However, we also consider the setting when we have a set of 'multiple' noisy labels. We compare against many weakly supervised methods, which are exactly methods that learn from multiple noisy labels.
>
> > It may be interesting if you could provide the cases when the proposed bounds fail or at least worse than the conventional one. It can tell the readers when should we use the proposed bound and when should not.
>
> We agree that the form of the two bounds are quite different and it is hard to say in general which is tighter. We note that our bound depends on the location of labeled points. Therefore, we believe that our bound is better when the labels are not drawn randomly. Also, as discussed in Section 3.2, the spectral bound is less tight when the graph is not well-clustered, while our bound also works in this setting.
>
> > Why does Table 2 not report the results of GCN, CLL and so on?
>
> Table 2 is included to address the notion of coverage of these methods. Most existing weakly supervised works (CLL, FS) do not attempt to improve coverage or expand the set of weak labels. Similarly, GCN does not incorporate weak supervision. Therefore, we mainly compare against standard LPA, Snorkel, and Liger (which is a weakly supervised method that does expand the set of weak labels).

---

### Official Review · Reviewer_s6EM · 2022-10-23

**Confidence:** 3
**Correctness:** 4
**Technical Novelty And Significance:** 4
**Empirical Novelty And Significance:** 3
**Recommendation:** 8

**Clarity, Quality, Novelty And Reproducibility:**

Although the math part requires knowledge about graph theory, such as conductance, the implication of theoretical results is clearly explained. For example, the strength of the new error bound against the existing one is explained using three examples shown in Figure 2.
The explanation in Section 4 could be improved. There are several new insights by extending the existing framework, but the impact compared with the existing works is a bit unclear to me.

It seems that the Dirichlet conductance from the recent study plays a vital role in Theorem 1. For the people who are not familiar with the study, some explanations/interpretations about the Dirichlet conductance would be helpful to understand this paper.

In the experiments, the average degree of the constructed Graph was set to 1. It seems a low value, and explanations for the value would improve the clarity of the paper.

In Theorem 4 in Appendix C, if explanations of the effect of t are provided, the clarity would be improved.

The error bound shown in this paper is novel. While the LPA framework in Section 4 is an extension of the existing framework, it is novel. It would be grateful if the authors could emphasize and clarify the contribution of the new framework while comparing the existing framework.

Regarding quality, the novel error bound provides more information than the existing spectral bounds. Moreover, the new LPA framework for incorporating information from multiple sources contributes to the study of LPA. It enables us to use weak labelers with LPA, and its usefulness was demonstrated in the experiments.

In Section 5 and Appendix G, the details of the experiments are provided. In addition, the code to replicate the experiments will be released publicly. The reproducibility is thus high.


**Strength And Weaknesses:**

##### Strengths
- The error bound that takes into account the geometric properties of the graph is novel.
- The proposed LPA with weak supervision allows us to use prior information from multiple sources.
- The effectiveness of the proposed method was demonstrated on various datasets.

##### Weaknesses
- The error bound depends on several parameters which need to be approximated in practice, as discussed in Section 6.

**Summary Of The Paper:**

This paper provides a "fine-grained" theory for the label propagation algorithm (LPA) in the presence of prior information. Unlike the existing theory, the new error bound takes into account the local geometric properties of the graph. The new error bound enables us to discriminate two geometrically different graphs with the same number of labeled nodes, while the existing bound cannot discriminate them. In addition to the new error bound, the existing framework for the LPA is extended to the setting where multiple sources of prior information are available. Finally, the superior performance of the proposed LPA with weak supervision was shown through numerical experiments.

**Summary Of The Review:**

This paper introduced the novel error bound for the label propagation algorithm. The difference between the existing and new bounds is clearly explained. The experimental results showed the empirical superiority of the new framework for LPA. The reproducibility seems high.

---

> ### Author Response · Authors · 2022-11-09
> **Response to Reviewer s6EM**
>
> Thank you for your review. We are pleased that you found our theoretical analysis to be “novel” and our proposed method to be “effective” on various datasets. We hope the following discussion addresses the questions you had.
>
> > In the experiments, the average degree of~ the constructed Graph was set to 1. It seems a low value, and explanations for the value would improve the clarity of the paper.
>
> We discuss our selection of $t$ (the value of the average degree) in Appendix G.2. We perform a hyperparameter sweep over values $t \in [1, 2, 5, 10, 100]$.
>
> > In Theorem 4 in Appendix C, if explanations of the effect of t are provided, the clarity would be improved.
>
> Sorry, this is a typo! Here, we have overloaded the meaning of $t$. $t$ here represents the # of times each vertex appears, where in our work $t$ represents the average degree of the graph. We change this variable to 'u' to improve the clarity.
>
> > It would be grateful if the authors could emphasize and clarify the contribution of the new framework while comparing the existing framework.
>
> The existing framework for learning from multiple sources of information as in the weak supervision is to aggregate them through a majority vote or Snorkel ((Ratner
> et al., 2016). We compared our new framework with the existing work in the experiment section. To the best of our knowledge, our framework is the first to incorporate multiple sources of information to label propagation.

---

### Official Review · Reviewer_Rb84 · 2022-10-23

**Confidence:** 4
**Correctness:** 3
**Technical Novelty And Significance:** 3
**Empirical Novelty And Significance:** Not applicable
**Recommendation:** 6

**Clarity, Quality, Novelty And Reproducibility:**


The proposed methods are sound, and the three aspects are pretty good.



**Details Of Ethics Concerns:**

No.

**Strength And Weaknesses:**

This paper provides some interesting results, but there are some questions needed to be answered.
1.	Definition 1 is confused in this paper, the subset S?
2.	What is the relationship between Theorem 1 and Theorem 2?
3.	In Theorem 2, you should give more details to introduce the generalization performance of the proposed method.
4.	In your experimental settings, the setting of 𝜆 should be more clear.
5.	Several mistakes need to be corrected. For example,
a.	“Proof. (Sketch) The key idea of our proof is to upper bound each E_i for ...”, and there is an error in mathematical notation in this sentence.
b.	“We note that this function…” This sentence also includes errors.
Therefore, I suggest that you should check your paper carefully to avoid similar mistakes.


**Summary Of The Paper:**

They propose a novel analysis of the classical label propagation algorithm with an error bound. They also propose a framework to incorporate multiple sources of noisy information.

**Summary Of The Review:**

In a word, the idea of the proposed model is useful to obtain a better performance, and the related theoretical support is sufficient. However, several detail issues are not clear in the proposed model, and some mistakes should be avoided.

---

> ### Author Response · Authors · 2022-11-09
> **Response to Reviewer Rb84**
>
> Thank you for your review. We are pleased that you found our theoretical analysis to be “novel” and to provide “interesting” results. We hope the following discussion addresses the questions you had.
>
> > Definition 1 is confused in this paper, the subset $S$?
>
> This is a typo! The definition is still valid, we have replaced $S$ with the text "any subset of nodes" in our updated version.
>
> > What is the relationship between Theorem 1 and Theorem 2?
>
> Theorem 2 is an existing bound that uses spectral analysis. We provide Theorem 2 to compare with our own analysis, which is given in Theorem 1. Section 3.2 is designed to compare these two different analyses, and demonstrates that our proposed bound is better in a concrete example.
>
> > In Theorem 2, you should give more details to introduce the generalization performance of the proposed method.
>
> As this is drawn from previous work (Belkin & Niyogi, 2004) to make a comparison, this is not our proposed method. We provide the original form of the theorem in Appendix C and details in section 3.2.
>
> > In your experimental settings, the setting of $\lambda$ should be more clear.
>
> We provide experiments on the WRENCH benchmark (Zhang et al., 2021), which comes with a standardized and fixed set of weak labels $\lambda$. Most of these weak labels come from simple engineered heuristics. We add an example of weak labeler $\lambda$ for the YouTube dataset in Appendix G.1, and we defer to the original benchmark paper for further details.
>
> > Several mistakes need to be corrected.
>
> Thank you for pointing these mistakes out! We have fixed them in our updated version.

---

### Official Review · Reviewer_H2Ah · 2022-10-24

**Confidence:** 3
**Correctness:** 4
**Technical Novelty And Significance:** 3
**Empirical Novelty And Significance:** Not applicable
**Recommendation:** 5

**Clarity, Quality, Novelty And Reproducibility:**


Clarity: Readability is fine. The paper is easy to follow. Illustrations help readers.

Quality: Technical quality is sufficiently good.

Novelty: Theoretical analysis contains a novel approach, while novelty on the multiple source extension is somewhat marginal.

Reproducibility: Experimental settings are sufficiently provided.

**Strength And Weaknesses:**

Strength:
- As the authors claim, the theoretical analysis provides a novel insight for label propagation. Differences from conventional analysis (Belkin & Niyoki 2004) described in Sec3.2 are interesting (e.g., conventional analysis does not reflect label positions in the graph). Overall, I think this analysis is valuable.
- The paper is easy to follow, and the proposed method is easy to implement. Relations with wide range of existing studies are discussed.
- Empirical evaluation shows good performance, compared with other recent weakly supervised models for the same setting.

Weaknesses:
- Theoretical justification of the choice of alpha_j in the multiple source extension is not fully clear (the main theoretical analysis is only for single weak supervision, and for multiple source setting, for which a novel model is proposed in this paper, a full theoretical analysis that includes the effect of the alpha estimation is not provided). In particular, the probabilistic model in Sec4.2 is a bit strange for me. Since y is binary label and h(x) is first defined as a map from X to [0,1], an interpretation of the model h_j \sim N(y,\sigma(x)) is unclear.
- The formulation of the proposed multiple source framework (3) itself is a bit straightforward. It is reasonable, but the technical novelty of this extension is not particularly strong. Further, compared with LPA+WL (single weak supervision), effectiveness of the multiple source extension is slight, empirically shown in Table1. The methodology of LPA+WL does not have particular novelty in my understanding (the objective function for the single supervision setting (1) is straightforward). In this sense, the performance superiority of the model newly proposed in this paper is not fully demonstrated.
- Although theoretical analysis is interesting, it is not reflected to the learning model and the algorithm, throughout the paper. Practical benefit of the theorem is not fully revealed.

Minor comments:
- 'set alpha_j(x_i) = 0 when h_j makes a correct prediction' should be 'alpha_j(x_i) = 1'?
- 'h_j \sim N(y,sigma(x))' should be 'h_j \sim N(y,sigma_j(x))'? The former definition seems to have a common variance across all h_j.
- Why w_ij does not exist in l(f) in Sec4.2?

**Summary Of The Paper:**

The topic of the paper is label propagation with an additional weak supervision that provides noisy predictions for all unlabeled points. The main contributions are two-folded: 1) providing a novel theoretical error bound of label propagation, and 2) proposing a multiple source extension. For 1), the analysis provides more detailed insight about the test error bound, compared with existing studies, and 2) proposes combining a set of different weak supervisions by adaptively weighting them based on their estimated accuracy. Performance superiority is empirically shown on four benchmark datasets.

**Summary Of The Review:**

For me, the paper is on the borderline, but I currently lean toward rejection. The theoretical analysis is interesting as written in 'strength', but it is only for the single supervision setting. On the other hand, the main methodological proposal of this paper is for the multiple information source setting. Nevertheless, the empirical results do not show clear performance superiority of the multiple source extension.

---

> ### Author Response · Authors · 2022-11-09
> **Response to Reviewer H2Ah**
>
> Thank you for your review. We are pleased that you found our theoretical analysis to be “valuable” and provide “novel insight” when compared to existing bounds. Thank you for picking up on multiple typos, which we have fixed in our updated version. We hope the following discussion addresses the questions you had.
>
> > choice of alpha_j in the multiple source extension is not fully clear (the main theoretical analysis is only for single weak supervision
>
> Although our theoretical analysis is for single weak supervision, we would like to point out that we provide a connection between single weak supervision and multiple sources of weak supervision in Appendix E. We show that there exists an alpha(x) that makes these two problems equivalent. This means we can easily generalize our theoretical analysis to the multiple weak supervision and provide guarantees for the extension.
>
> > In particular, the probabilistic model in Sec4.2 is a bit strange for me.
>
> This model serves as a potential interpretation of the multiple sources of information and provides us with a natural framework to develop a weighting scheme. The choice of normal distribution h_j \sim N(y, \sigma_j(x)) leads to a likelihood function that we can efficiently solve through label propagation.  While we could model h_j \sim Bernoulli(p_j), it is not clear how to solve this problem efficiently. We explore different schemes for determining alpha, including an accuracy-based and boosting-based approach (Appendix F), which provide alternative interpretations of the multiple sources of weak supervision.
>
> > The methodology of LPA+WL does not have particular novelty in my understanding (the objective function for the single supervision setting (1) is straightforward).
>
> The analysis is the novel component of our work. We analyze the most natural extension of LPA to handle weak supervision or noisy labels.
>
> > The formulation of the proposed multiple source framework (3) itself is a bit straightforward. It is reasonable, but the technical novelty of this extension is not particularly strong.
>
> While our formulation to multiple sources is a natural extension, it is a novel setting and has not been considered before. We also provide multiple weighting schemes to incorporate these multiple sources of information, and we believe they are also novel contributions.
>
> >'set alpha_j(x_i) = 0 when h_j makes a correct prediction' should be 'alpha_j(x_i) = 1'?
> 'h_j \sim N(y,sigma(x))' should be 'h_j \sim N(y,sigma_j(x))'? The former definition seems to have a common variance across all h_j.
> Why w_ij does not exist in l(f) in Sec4.2?
>
> Thank you for pointing this out! We will correct this typo.

---

### Author Response · Authors · 2022-11-09
**General Response**

We would like to thank all the reviewers for their efforts in providing thoughtful and attentive examinations of our work. We are glad to see that the reviewers highlighted a number of strengths, including:

* Our novel theoretical analysis (H2Ah, Rb84, s6EM, xkhm)
* Our comparison to existing spectral bounds is “interesting” (H2Ah) and shows  “superiority” on some examples (xkhm)
* Empirical evaluation shows “good performance” (H2Ah, xkhm) and is effective on various datasets (s6EM)
* High reproducibility (s6EM) and easy to implement (H2Ah)

We respond to individual reviewer comments in the individual threads. Thank you again for your hard work and consideration.

---

### Decision · Program_Chairs · 2023-01-20

**Decision:**

Accept: poster

**Justification For Why Not Higher Score:**

Since the proposed algorithm itself is rather straightforward, innovativeness is not high enough to recommend this work to spotlight or oral.

As several reviewers pointed out, explanations in Section 4 are not clear enough and thus I suggest the authors to revise it based on the feedback from the reviewers.



**Justification For Why Not Lower Score:**

I recommend the acceptance of this paper due to its technical soundness and empirical usefulness.

**Metareview: Summary, Strengths And Weaknesses:**

Summary:
This paper provides a novel theoretical analysis of the classical label propagation algorithm and establishes an error bound. The authors also incorporate multiple sources of noisy information. Experiments demonstrate the usefulness of the label propagation algorithm with weak supervision.

Strength:
Theory is insightful and empirical performance is promising.

Weakness:
The proposed algorithm is rather straightforward.


**Note From Pc:**

if the above contains the word "oral" or "spotlight" please see: "oral" presentation means -> notable-top-5% and "spotlight" means -> notable-top-25%. As stated in our emails, we are disassociating presentation type from AC recommendations